# MEASURING SCARCITY–COMPLEXITY COLLISION IN LANGUAGE MODEL ESTIMATION

## ABSTRACT

Formal languages are increasingly used to analyze limitations of language–model architectures, via properties of their defining automata (e.g., number of states, transition weights, or out-degree at a state). Understanding why a neural language model struggles to learn a given property requires separating two cases: **scarcity**—the property is underrepresented in the data due to low likelihood—versus **complexity**—the task being too sample-demanding for the chosen learner. In the former case, increasing or resampling training data can help; in the latter, changes to the architecture or training may be needed. Evaluating a property "on its own" typically involves marginalizing over a family of languages that exhibit it, which can introduce selection bias: some family members may be systematically overrepresented among samples with the property, confounding correlational analyses. Indeed, most existing investigations relate corpus statistics to performance, but such correlations do not identify causal effects. We introduce a causal framework for assessing learnability in probabilistic formal languages. Using an efficient, controlled sampling procedure for regular languages (faster by a magnitude of $N^2$ where $N$ is the number of tracked event occurrences), we *intervene on event frequencies by replacing the unconstrained sampling policy with a controlled one*, leaving the automaton's topology and weights unchanged, to test whether a property looks difficult because it is rare or because it is genuinely demanding for the learner. Our main contribution is theoretical: the framework and controlled sampling procedures. We illustrate these in three example case studies with LSTM and Transformer learners, where conclusions under correlational evaluation can invert once causal frequency interventions are applied.

## 1 INTRODUCTION

When a neural language model (LM) fails to learn a pattern, it matters whether the difficulty stems from *scarcity*—too little evidence for the relevant phenomenon—or from a task that is *more sample-demanding for the chosen learner under our settings*. This distinction shapes how we interpret empirical results: in the former case, increasing or resampling training data can help; in the latter, changes to the architecture or training may be needed. We lack systematic methods to diagnose these cases, and theoretical bounds for what Transformer LMs can learn remain limited—existing results mostly concern recognizers rather than language models as generators (Yang et al., 2024). In practice, studies typically *sample tasks* and relate *task properties* (e.g., number of states, entropy) to *learner performance* across tasks.

Studying these questions directly in natural language is difficult because the data-generation process is not controllable. Formal languages provide controlled testbeds: we can sample *language models* induced by probabilistic finite-state automata (PFSAs) and vary their properties systematically. Prior work benchmarks learners against structural, grammatical, and distributional complexity measures (Michalenko et al., 2019; Valvoda et al., 2022; Van der Poel et al., 2025; Borenstein et al., 2024; Svete et al., 2024a; Delétang et al., 2023; Someya et al., 2025). For illustration, reports under vanilla training find that Transformers can lag LSTMs on parity-like signals (Hahn & Rofin, 2024) (see Fig. 1 for a concrete parity-augmented testbed). These studies are *correlational*: they draw many tasks from a family and analyze post-hoc associations between task properties and performance. Such associations can conflate rarity with difficulty. We use *scarcity–complexity collision* to describe the entanglement of these two drivers of learnability.

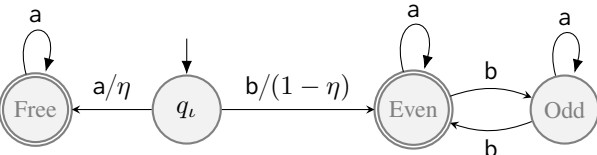

Figure 1: A DFSA recognizing the language aa* ∪ ba*(ba*ba*)*. The transitions left of $q_\iota$ correspond to the star-free language aa* and the ones to the right to the more complex parity language, requiring counting modulo two.

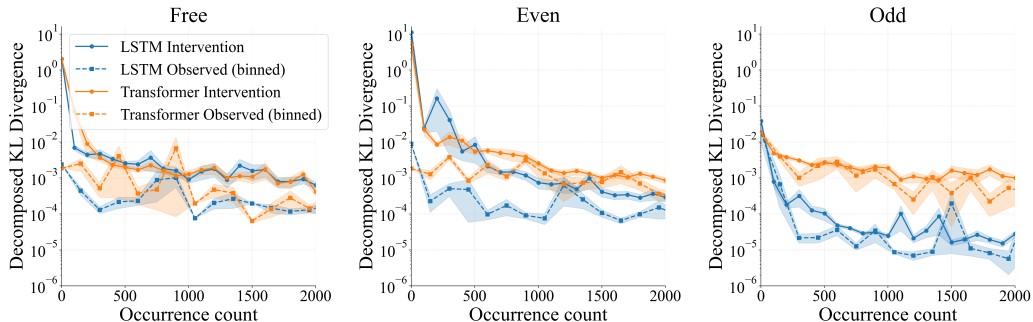

Figure 2: Interventional and correlational estimates for the automaton in Fig. 1. Panels report *state-wise* decomposed KL divergence (lower better) versus the training-set occurrence count, isolating learnability of individual properties. Note the alignment for *Odd*, in contrast to *Free* and *Even*

To address the scarcity–complexity question, we adopt a *causal* study that controls both the amount of data and its composition. Concretely, we intervene on specific dataset properties—such as the frequency of particular symbols, states, or transitions—while holding other factors fixed. Executing such interventions is infeasible in natural language, where the generating mechanism is unknown, but formal languages offer precise control. We therefore work with *language models* induced by probabilistic finite-state automata (PFSAs), following prior evaluation setups that use PFSA-induced tasks to probe learners (Lake & Baroni, 2018; Ruis et al., 2020; Hupkes et al., 2020, *inter alia*). Prior studies typically draw many tasks from a family and analyze post-hoc associations between task properties and performance; this *correlational* protocol can suffer from selection bias, some PFSA instances are more likely among samples that exhibit a given property, conflating rarity with difficulty. In contrast, we replace the unconstrained sampling policy with a controlled one that targets event frequencies, leaving PFSA topology and weights unchanged. This lets us vary *complexity* by sampling different PFSA topologies, and vary *scarcity* by adjusting the data-sampling procedure for a fixed PFSA.

We develop a causal LM evaluation framework (Pearl et al., 2016) with two components: *(i)* a causal graphical model relating a PFSA, properties of datasets sampled from it, and LM performance when *trained* on those datasets; and *(ii)* an efficient (from $\mathcal{O}(|Q|^3 N^4)$ to $\mathcal{O}(|Q|^3 N^2)$) controlled sampler that fixes how often specified properties (symbols, states, transitions) appear—e.g., exact counts or at-least-once constraints. We *illustrate* the setup with three case studies using Transformer (Vaswani et al., 2017) and LSTM (Hochreiter & Schmidhuber, 1997) learners, and evaluate learnability at the level of states, symbols, and transitions. In these illustrations, increasing the frequency of rare features can improve learnability, but not always; moreover, conclusions drawn from correlational evaluation can *invert* under causal frequency interventions. This makes it possible to diagnose whether observed difficulty reflects *scarcity* or *sample demand* in this setting.

## 2 LANGUAGE MODELS, SCARCITY AND COMPLEXITY

We now formalize what we have so far informally referred to as scarcity and complexity. An **alphabet** $\Sigma$ is a finite non-empty set of **symbols**. The **Kleene closure** $\Sigma^*$ is the set of all strings of symbols

from $\Sigma$. A **language model** $p$ is a probability distribution over $\Sigma^*$. The **prefix probability**[1] $\overrightarrow{p}(\boldsymbol{\sigma})$ of a string $\boldsymbol{\sigma} \in \Sigma^*$ is the probability mass of strings starting with $\boldsymbol{\sigma}$ under $p$; see Eq. (1). We also define the **conditional** prefix probability in Eq. (2).

$$\overrightarrow{p}(\boldsymbol{\sigma}) \stackrel{\text{def}}{=} \sum_{\boldsymbol{\sigma}' \in \Sigma^*} p(\boldsymbol{\sigma}\boldsymbol{\sigma}') \qquad (1) \qquad \overrightarrow{p}(\boldsymbol{\sigma}' \mid \boldsymbol{\sigma}) \stackrel{\text{def}}{=} \frac{\overrightarrow{p}(\boldsymbol{\sigma}\boldsymbol{\sigma}')}{\overrightarrow{p}(\boldsymbol{\sigma})} \quad \text{if } \overrightarrow{p}(\boldsymbol{\sigma}) > 0 \text{ else } 0. \qquad (2)$$

Many LMs define $p(\boldsymbol{\sigma})$ as $p(\boldsymbol{\sigma}) \stackrel{\text{def}}{=} \overrightarrow{p}(\text{EOS} \mid \boldsymbol{\sigma}) \prod_{t=1}^{|\boldsymbol{\sigma}|} \overrightarrow{p}(\sigma_t \mid \boldsymbol{\sigma}_{<t})$, where $\text{EOS} \notin \Sigma$ is a designated end-of-string symbol. We denote neural LMs with $p_{\boldsymbol{\theta}}$, and define the conditional distributions with some parameters $\boldsymbol{\theta}$. Suppose we estimate $\boldsymbol{\theta}$ on a dataset $\mathcal{D}$ sampled from a ground truth LM $p$. It could be that the individual conditional prefix probabilities $\overrightarrow{p_{\boldsymbol{\theta}}}(\cdot \mid \boldsymbol{\sigma})$ of the LM $p_{\boldsymbol{\theta}}$ are not learned well for some prefixes, i.e., $\overrightarrow{p_{\boldsymbol{\theta}}}(\cdot \mid \boldsymbol{\sigma})$ may not be close to $\overrightarrow{p}(\cdot \mid \boldsymbol{\sigma})$. It is difficult to know *why* $\overrightarrow{p_{\boldsymbol{\theta}}}(\cdot \mid \boldsymbol{\sigma})$ is poorly learned. This could be due to **scarcity** of the specific pattern in the training data due to the low likelihood of observing it. However, it could also be that $\overrightarrow{p}(\cdot \mid \boldsymbol{\sigma})$ is something inherently hard to learn—either because representing the distribution is a computationally hard problem, because the chosen architecture cannot represent it well, or because the learning dynamics prevent it from being learned. Both explanations are possible when estimating $\boldsymbol{\theta}$ from i.i.d.-sampled data.

As a concrete example, consider string representations of graphs and the language consisting of those that encode a path visiting each node exactly once. Training a model to recognize such strings is tantamount to solving the NP-hard Hamiltonian path problem. While the amount and composition of training data may affect performance in practice, we expect the task to require exponential computation in the size of the graph (assuming P $\neq$ NP), regardless of how much data is available. Compare this to learning the language defined by the PFSA in Fig. 1, which defines an LM whose support is the union of aa* and ba*(ba*ba*)*. The former should be easy to learn for neural LMs, while the second, requiring counting modulo two, is known to be hard for Transformers (Hahn & Rofin, 2024). A low value of the transition probability $\eta$ would, however, result in few aa*-sequences, possibly leading to the false conclusion that this part of the language is hard to learn. Similarly, if $\eta$ is large, we might assume that the harder part of the language is difficult for the wrong reason. Thus, distinguishing the *rare-but-learnable* events from *hard to learn* becomes difficult when both are entangled in a single dataset.

We illustrate this with a simple setup by sampling training data from the machine in Fig. 2 and plotting the learnability of each state as a function of how often the state is visited (note the $\log$-scaled $y$-axis). First, we see a clear difference under interventional and observational distributions for the accepting states (Free and Even), showing that intervening, such as forcing otherwise high-frequency events to occur rarely, leads to a significant increase in the KL divergence for accepting states. This shows that the observed learnability of a language feature is not necessarily indicative of its inherent learnability when controlling for confounding variables. Second, for the Even state, we see that LSTMs trained on intervened data eventually catch up with the observational ones, but only after a large number of samples. At the same time, the difference to the Transformer remains, even as we increase the number of samples—reflecting the known fact that Transformers are not able to learn the parity language (Hahn & Rofin, 2024). We see that the 'Odd' state is easier to learn than the others for the LSTM, while such a distinction can not be seen for the Transformer.

## 3 A CAUSAL MODEL FOR LEARNABILITY

Understanding whether poor model performance stems from task complexity or data scarcity due to low probabilities requires moving beyond correlation to examine causal relationships. In this section, we develop a causal framework based on Pearl's do-calculus (Pearl et al., 2016) to isolate the effects of data properties on language model performance.[2]

Standard correlational analysis examines how model performance varies with dataset properties, but this approach suffers from a fundamental limitation: Datasets with certain properties might be generated more frequently by certain types of language models. For example, a dataset with many

---

[1]We caution the reader that prefix probabilities do *not* define a distribution over $\Sigma^*$, as they do not sum to 1.

[2]In App. A, we provide the necessary background in formal language theory and graphical causal models.

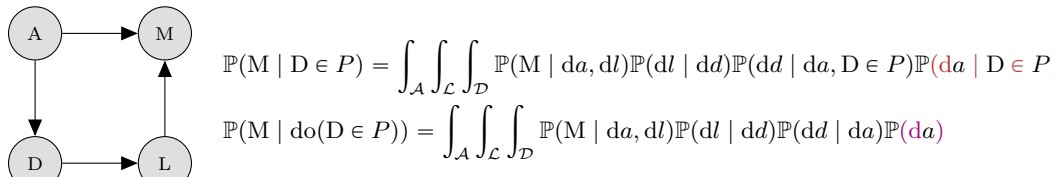

$$\mathbb{P}(\mathrm{M} \mid \mathrm{D} \in P) = \int_{\mathcal{A}} \int_{\mathcal{L}} \int_{\mathcal{D}} \mathbb{P}(\mathrm{M} \mid \mathrm{d}a, \mathrm{d}l)\mathbb{P}(\mathrm{d}l \mid \mathrm{d}d)\mathbb{P}(\mathrm{d}d \mid \mathrm{d}a, \mathrm{D} \in P)\mathbb{P}(\mathrm{d}a \mid \mathrm{D} \in P)$$

$$\mathbb{P}(\mathrm{M} \mid \mathrm{do}(\mathrm{D} \in P)) = \int_{\mathcal{A}} \int_{\mathcal{L}} \int_{\mathcal{D}} \mathbb{P}(\mathrm{M} \mid \mathrm{d}a, \mathrm{d}l)\mathbb{P}(\mathrm{d}l \mid \mathrm{d}d)\mathbb{P}(\mathrm{d}d \mid \mathrm{d}a)\mathbb{P}(\mathrm{d}a)$$

Figure 3: **Left:** Graphical causal model for evaluating the effect of intervening on the dataset D on model performance M, given a ground truth language model A and trained model L. **Right:** Conditioning (top) induces $\mathbb{P}(a \mid \mathrm{D} \in P)$; intervening (bottom) restores the marginal $\mathbb{P}(a)$.

occurrences of a particular symbol is more likely to come from a language model that frequently generates that symbol, regardless of the complexity of the language. Causal analysis addresses this limitation by actively intervening on the data generation process. An *intervention* artificially modifies the sampling process to ensure specific dataset properties, regardless of how likely these properties would occur naturally.

We formalize our approach using the causal graphical model shown in §3. This model represents the process of: *(i)* sampling a ground-truth language model A, *(ii)* generating a dataset D from this language model, *(iii)* training a neural LM L on this dataset, and *(iv)* evaluating the trained model's performance M. In our formal model, each of these components corresponds to a random variable in a probability space.[3] In §4, we focus on a specific specification of this model.

**Intervening vs. Conditioning** We define a property $P$ of a dataset as an *event* over $\mathcal{D}$. For example, the property "the dataset contains exactly $N$ occurrences of symbol $a$." The key to our causal analysis is distinguishing:

(1) **Conditioning.** Evaluate $\mathbb{P}(\mathrm{M} \mid \mathrm{D} \in P)$, which introduces dependence between A and D (the term $\mathbb{P}(a \mid \mathrm{D} \in P)$ in §3). Models more likely to produce datasets with $P$ may be overrepresented.

(2) **Intervention.** Evaluate $\mathbb{P}(\mathrm{M} \mid \mathrm{do}(\mathrm{D} \in P))$ by modifying the data generation to satisfy $P$, preserving the original distribution $\mathbb{P}(a)$ of ground-truth models.

In practice, we implement interventions with a sampling method (§4) that generates datasets with specified properties from any PFSA. We then estimate $\mathbb{E}[\mathrm{M} \mid \mathrm{do}(\mathrm{D} \in P)]$ via Monte Carlo sampling.

**Causal estimands.** For any dataset property $P$ (e.g., "exactly $N$ occurrences of $a$ in a corpus of $K$ strings"), we define the interventional and observational performance, and their difference, as

$$\mu_{\mathrm{int}}(P) = \mathbb{E}\left[\mathrm{M} \mid \mathrm{do}(\mathrm{D} \in P)\right], \quad \mu_{\mathrm{obs}}(P) = \mathbb{E}\left[\mathrm{M} \mid \mathrm{D} \in P\right], \quad \Delta_{\mathrm{causal}}(P) = \mu_{\mathrm{int}}(P) - \mu_{\mathrm{obs}}(P), \tag{3}$$

which isolates the effect of enforcing $P$ on performance while preserving the original distribution of A (cf. Fig. 3). Because $\mathrm{do}(\mathrm{D} \in P)$ is implemented exactly by our controlled sampler (§4), $\mu_{\mathrm{int}}(P)$ is identified from interventional data. We estimate both $\mu_{\mathrm{int}}$ and $\mu_{\mathrm{obs}}$ by Monte Carlo over (i) datasets from the respective distributions and (ii) independent training runs of L; we report means and standard errors across seeds. In our experiments, M is instantiated as a decomposed divergence local to the point of intervention (definition in App. D).

The next section details efficient controlled sampling from the intervened on distribution.

## 4 SAMPLING UNDER EVENT CONSTRAINTS

§3 discusses how to evaluate the effects of dataset properties on model performance. Here, we develop efficient algorithms for sampling strings from deterministic formal automata (DPFSAs) under interventions on the properties of the dataset, i.e., from $\mathbb{P}(d \mid \mathrm{A} = \mathcal{A}, \mathrm{do}(\mathrm{D} \in P))$.

---

[3]We include the randomness of the trained model $l$ given the dataset $d$ (initialization, shuffling, etc.); we omit $\sigma$-algebras here and refer to App. A.

**Interventions of interest.** To develop efficient algorithms, we focus on properties $P$ of a particular type: We require the strings in the dataset to traverse a set of transitions $\mathcal{T}$ in the PFSA exactly $N \in \mathbb{N}$ times. This formulation is natural as well as expressive: Besides controlling the appearance of particular (sets of) transitions, it also allows controlling *(i)* the number of times a symbol a appears in the dataset (by choosing $\mathcal{T}$ to be all a-labeled transitions), and *(ii)* the number of times a state $q$ appears in the dataset as a latent variable (by choosing $\mathcal{T}$ to be all transitions with $q$ as the source state).

To illustrate sampling from $\mathbb{P}(d \mid \mathrm{A} = \mathcal{A}, \mathrm{do}(\mathrm{D} \in P))$, suppose that $P$ takes the form that a symbol a has to occur $N \in \mathbb{N}$ times. One can sample from the interventional distribution as follows. First, determine the number of strings, $K$, in the dataset, and then sample the number of times a has to appear in each of the $K$ strings. This reduces the problem to the following: For $i \in \mathbb{N}$, sample a string from $p(\boldsymbol{\sigma} \mid \boldsymbol{\sigma}$ has $i$ symbols a$)$, where $p$ is a PFSA-induced LM. Simple rejection sampling (sampling a large dataset and only keeping the strings that satisfy the constraints) would be highly impractical. Using standard algorithms from automata theory, a more efficient approach is as follows: *(i)* For every $i \in \{0, \ldots, N\}$, construct an FSA $\mathcal{A}_{\mathrm{a},i}$ that recognizes the language of all strings with exactly $i$ a's, *(ii)* intersect $\mathcal{A}_{\mathrm{a},i}$ with $\mathcal{A}$ to obtain the automaton $\mathcal{A}_{\mathrm{a},i} \cap \mathcal{A}$ that recognizes the language of all strings with exactly $i$ a's that are also accepted by $\mathcal{A}$, *(iii)* renormalize $\mathcal{A}_{\mathrm{a},i} \cap \mathcal{A}$'s weights to obtain a PFSA using the weight pushing algorithm (Mohri, 2009) to allow sampling. Here, intersection results in a PFSA with $\mathcal{O}(|Q|i)$ states, and weight pushing runs in cubic time with respect to the number of states, so the overall runtime of this procedure for all $i$ would be $\mathcal{O}(|Q|^3 N^3 N) = \mathcal{O}(|Q|^3 N^4)$ since intersection and weight pushing have to be done for each $i$ separately. In the following, we improve on this by giving an algorithm that runs in $\mathcal{O}(|Q|^3 N^2)$ time.

## 4.1 The Binning Semiring

To efficiently encode string constraints for sampling, we introduce a new algebraic structure—the **binning semiring** $[\mathbb{K}]$. The elements of the semiring are formal polynomials of the form

$$\sum_{i=0}^{N} a_i x^i \tag{4}$$

where $a_i \in \mathbb{K}$ for some **base semiring** $\mathbb{K}$ and $x$ is a variable. The semiring addition $\oplus$ is defined element-wise with the unit $\boxed{0} \stackrel{\text{def}}{=} \sum_{i=0}^{N} \mathbf{0} x^i$. Multiplication is defined as

$$\left(\sum_{i=0}^{N} a_i x^i\right) \otimes \left(\sum_{i=0}^{N} b_i x^i\right) \stackrel{\text{def}}{=} \sum_{i=0}^{N} \left(\sum_{j=0}^{i} a_j \otimes b_{i-j}\right) x^i + \left(\sum_{\substack{i,j=0 \\ i+j \geqslant N}}^{N} a_i \otimes b_j\right) x^i \tag{5}$$

with the unit $\boxed{1} \stackrel{\text{def}}{=} \mathbf{1} x^0 + \sum_{i=1}^{N} \mathbf{0} x^i$. The binning semiring is a *monoid semialgebra* over a *cyclic monoid*. This gives it rich algebraic properties; we discuss the details in App. B.1.

**Notation.** Formalization of the binning semiring as a formal power series is convenient due to the connection to monoid semialgebras. However, it makes some of the notation slightly cumbersome. In the following, we therefore think of the elements in the binning semiring as *vectors* of the form $\boldsymbol{v} = \begin{pmatrix} a_0 & a_1 & \cdots & a_N \end{pmatrix}^\top$ where $a_i$ are the coefficients of the polynomial in Eq. (4). We thus access the coefficients of the polynomial as $v_i$ for $i \in \{0, \ldots, N\}$.

**Time complexity.** The time complexities of the operations in the binning semiring depend on the order $N$. Assuming $\oplus$ and $\otimes$ run in constant time, $\boxed{\oplus}$ runs in time $\mathcal{O}(N)$. The $\boxed{\otimes}$ operation (a convolution) runs in time $\mathcal{O}(N^2)$. In the special case of real-weighed WFSAs, however, we can implement $\boxed{\otimes}$ using the fast Fourier transform (FFT) in time $\mathcal{O}(N \log N)$.[4]

## 4.2 Binning Automata

We now describe how to **lift** a PFSA $\mathcal{A}$ to a WFSA over the binning semiring.

---

[4]Despite the faster implementation of $\boxed{\otimes}$ with FFT, we do not adopt it due to numerical instability (underflow for large values of $N$). Instead, we use the log semiring for numerical stability.

**Definition 4.1** (Event and Lifting Functions). Let $\mathcal{A} = (Q, \Sigma, \delta, \lambda, \rho)$ be a WFSA and $N \in \mathbb{N}$. An **event function** is a tuple $\phi = (\phi_\lambda, \phi_\delta, \phi_\rho)$, where $\phi_\delta \colon Q \times \Sigma \times Q \to \{0, \dots, N\}$ and $\phi_\lambda, \phi_\rho \colon Q \to \{0, \dots, N\}$. For $\phi_f$ and $i \in \{0, \dots, N\}$, we define the **lifting function** $\mathcal{L}_f \colon X \to [\mathbb{K}]$:

$$\mathcal{L}_f(x)_i \overset{\text{def}}{=} \begin{cases} f(x) & \textbf{if } i = \phi_f(x) \\ \mathbf{0} & \textbf{otherwise} \end{cases} \tag{6}$$

Informally, the lifting function takes a (transition, initial, or final) weight in the original WFSA and inserts it into the $\phi_f(x)^{\text{th}}$ position of the otherwise $\mathbf{0}$-valued lifted weight $\in [\mathbb{K}]$.

**Definition 4.2** (Binning Automaton). Let $\mathcal{A} = (Q, \Sigma, \delta, \lambda, \rho)$ be a WFSA. The **binning automaton** of $\mathcal{A}$ is the WFSA $\mathcal{A}_\phi = (Q, \Sigma, \delta_\phi, \lambda_\phi, \rho_\phi)$, where $f_\phi(x) = \mathcal{L}_{\phi_f}(x)$ for $f \in \{\delta, \lambda, \rho\}$.

We can count the number of times an event $\phi$ occurs on a path $\boldsymbol{\pi} = q_0 \xrightarrow{a_1/w_1} \cdots \xrightarrow{a_m/w_m} q_m$ as

$$|\boldsymbol{\pi}|_\phi \overset{\text{def}}{=} \phi_\lambda(q_0) + \sum_{i=1}^{m} \phi_\delta(q_{i-1}, a_i, q_i) + \phi_\rho(q_m). \tag{7}$$

Similarly, for $\mathcal{K} \subset \mathbb{N}$ and a multiset of paths $\mathcal{P} = \{\boldsymbol{\pi}_k\}_{k \in \mathcal{K}}$, we write $|\mathcal{P}|_\phi \overset{\text{def}}{=} \sum_{k \in \mathcal{K}} |\boldsymbol{\pi}_k|_\phi$. For path weights in $\mathcal{A}_\phi$, we will subscript the path weight function: $\boldsymbol{w}_\phi(\boldsymbol{\pi}) \in [\mathbb{R}]$ for $\boldsymbol{\pi} \in \Pi(\mathcal{A}_\phi)$.

### 4.2.1 Constrained Sampling

We use the binning semiring to implement a faster constrained sampling algorithm by *sharing* computations among different values of $i$. When the original WFSA is a PFSA, the $i^{\text{th}}$ coefficient of a transition weight in the binning automaton holds the probability of $i$ occurrences of $\phi$ occurring on that transition. This gives the interpretation that the probabilities of the original PFSA are *binned* by the number of events. Running weight pushing on the new automaton gives us probability distributions specific to each $i$ that tell us, from any given state, the probability that each action generates exactly $i$ events in the future, leading to a straightforward sampling algorithm. We formalize this in intuition below. In the following, $\mathcal{A}$ is always a DPFSA, $\phi$ an event function, and $\mathcal{A}_\phi$ $\mathcal{A}$'s binning automaton. We first formalize the intuition that $\boldsymbol{w}_\phi(\boldsymbol{\pi})$ is a one-hot encoding whose non-zero coefficient encodes the number of occurrences of an event $\phi$. Proofs of all theorems in this section are included in App. C.

**Theorem 4.1** (Path Weight Interpretation). *For $\boldsymbol{\pi} \in \Pi(\mathcal{A}_\phi)$ and $i \in \{0, \dots, N-1\}$, we have:*

$$\overline{\boldsymbol{w}}_\phi(\boldsymbol{\pi})_i = \begin{cases} \overline{\boldsymbol{w}}(\boldsymbol{\pi}) & \textbf{if } i < N \text{ and } |\boldsymbol{\pi}|_\phi = i \text{ or } i = N \text{ and } |\boldsymbol{\pi}|_\phi \geqslant N \\ 0 & \textbf{otherwise} \end{cases} \tag{8}$$

*where $\overline{\boldsymbol{w}}(\boldsymbol{\pi})$ is the inner path weight in $\mathcal{A}$.*

Next, we formalize the intuition that summing path weights results in a vector that encodes the probabilities of sampling specific numbers of occurrences. To formalize this, we introduce the random variable $\boldsymbol{\Pi}(q)$ distributed as

$$\mathbb{P}(\boldsymbol{\Pi}(q) = q \xrightarrow{a_1/w_1} \cdots \xrightarrow{a_m/w_m} q_m) \overset{\text{def}}{=} w_1 \otimes \cdots \otimes w_m \otimes \rho(q_m). \tag{9}$$

**Theorem 4.2** (Backward Weight Interpretation). *For $q \in Q$, we have:*

$$\boldsymbol{\beta}_\phi(q)_i = \begin{cases} \mathbb{P}(|\boldsymbol{\Pi}(q)|_\phi = i) & \textbf{if } i < N \\ \mathbb{P}(|\boldsymbol{\Pi}(q)|_\phi \geqslant N) & \textbf{otherwise} \end{cases} \tag{10}$$

In words, Thm. 4.2 tells us that the probability of seeing exactly (or at least) $i$ occurrences of $\phi$ in a random path starting in $q \in Q$ is the value of the $i^{\text{th}}$ element of $q$'s backward weight. Computing backward weights in a general PFSA with cycles requires summation over infinitely many paths. This can be done efficiently with *asteration*—taking the Kleene closure of the set of paths. We derive the closed-form solution for asteration in the binning semiring applied to PFSAs in App. B.2.

We use the binning automaton to sample from DPFSAs under event-counting interventions of the form $|\mathcal{P}| = K$ and $|\mathcal{P}|_\phi = N$ for $\mathcal{P} = \{\boldsymbol{\pi}_k\}_{k \in \mathcal{K}}$, $\mathcal{K} \subset \mathbb{N}$, and $N \in \mathbb{N}$. We sample $\mathcal{P}$ in *two steps*: First, knowing $|\mathcal{K}|$, we sample the number of events $|\boldsymbol{\pi}_i|_\phi$ for all $\boldsymbol{\pi}_i \in \mathcal{P}$. Then, we sample the paths $\boldsymbol{\pi}_k$ for $k \in \mathcal{K}$ under the determined string-level event constraints.

**Theorem 4.3** (Counting in a Set). *Denoting with* $\boldsymbol{Z} \stackrel{\text{def}}{=} \boldsymbol{\beta}_\phi(q_\iota) = \sum_{\boldsymbol{\pi} \in \Pi(\mathcal{A}_\phi)} \boldsymbol{w}_\phi(\boldsymbol{\pi})$ *the sum of all path weights in the binning automaton* $\mathcal{A}_\phi$, *we have:*

$$(\boldsymbol{Z}^{\otimes K})_i = \begin{cases} \mathbb{P}(|\mathcal{P}|_\phi = i) & \textbf{if } i < N \\ \mathbb{P}(|\mathcal{P}|_\phi \geqslant N) & \textbf{otherwise} \end{cases} \tag{11}$$

**Theorem 4.4** (Sampling Path Event Counts (1)). *We have that*

$$\mathbb{P}(|\boldsymbol{\pi}_k|_\phi = i \mid |\mathcal{P}|_\phi = N, |\{\boldsymbol{\pi}_j\}_{j<k}|_\phi = m) = \frac{(\boldsymbol{Z}^{\otimes K-k-1})_{N-m-i}}{(\boldsymbol{Z}^{\otimes K-k})_{N-m}} \boldsymbol{Z}_i \tag{12}$$

We derive the expression for $\mathbb{P}(|\boldsymbol{\pi}_k|_\phi \geqslant i \mid |\mathcal{P}|_\phi \geqslant N, |\{\boldsymbol{\pi}_j\}_{j<k}|_\phi = m)$ in Thm. C.1. Moreover, to be able to control how many individual strings contain at least one occurrence of $\phi$, we develop a method to sample from $\mathbb{P}(|\boldsymbol{\Pi}_k(q_\iota)|_\phi \geqslant i \mid |\mathcal{P}|_{\mathbb{1}\{\phi\}} = M, |\{\boldsymbol{\pi}_j\}_{j<k}|_{\mathbb{1}\{\phi\}} = m)$ where $|\mathcal{P}|_{\mathbb{1}\{\phi\}} \stackrel{\text{def}}{=} \sum_{k \in \mathcal{K}} \mathbb{1}\{|\boldsymbol{\pi}_k|_\phi \geqslant i\}$ in Thm. C.2.

Thm. 4.4 tells us how many events we should ask for in each path when sampling under intervention. To sample strings with the determined number of occurrences of $\phi$, i.e., from $\mathbb{P}(\boldsymbol{\Pi}(q_\iota) = \boldsymbol{\pi} \mid |\boldsymbol{\Pi}(q_\iota)|_\phi = n)$, we define another, larger, PFSA $\mathcal{A}' = (\Sigma, Q', \delta', \lambda', \rho')$ as

    (i) $Q' = Q \times \{0, \ldots, N-1\}$,

    (ii) $\delta': ((q,i), \mathsf{a}, (q',i)) \mapsto \frac{1}{\boldsymbol{Z}(q)_{N-1-i}} \big(\mathcal{L}_{\phi_\delta}(q \xrightarrow{\mathsf{a}/w} q') \otimes \boldsymbol{\beta}_\phi(q')\big)_{N-1-i}$,

    (iii) $\lambda': (q,i) \mapsto \mathbb{1}\{q = q_\iota, i = N-1\},$[5]

    (iv) $\rho': (q,i) \mapsto \frac{1}{\boldsymbol{Z}(q)_{N-1-i}} \mathcal{L}_\rho(q)_{N-1-i}$.

Here, $\boldsymbol{Z}(q) \stackrel{\text{def}}{=} \boldsymbol{\beta}_\phi(q)$. Intuitively, $\mathcal{A}'$ being in state $(q,i)$ means that $N-1-i$ occurrences of $\phi$ are still required for the string generation to be complete. We will denote with $\boldsymbol{\Pi}'(q_\iota)$ the random variable distributed analogously to $\boldsymbol{\Pi}(q_\iota)$, but in the new PFSA $\mathcal{A}'$.

**Theorem 4.5** (Constrained String Sampling (1)). *We have that*

$$\mathbb{P}(\boldsymbol{\Pi}(q_\iota) = \boldsymbol{\pi} \mid |\boldsymbol{\Pi}(q_\iota)|_\phi = N-1) = \mathbb{P}(\boldsymbol{\Pi}'(q_\iota) = \boldsymbol{\pi}). \tag{13}$$

We consider the analogous case of sampling from $\mathbb{P}(\boldsymbol{\Pi}(q_\iota) = \boldsymbol{\pi} \mid |\boldsymbol{\Pi}(q_\iota)|_\phi \geqslant n)$ in Thm. C.3. To summarize, Thms. 4.4 and 4.5 describe sampling from $\mathcal{A}$ to get a specific number of events in a dataset. Thm. 4.4 tells us how many occurrences of $\phi$ each string should contain and Thm. 4.5 tell us how to sample said strings. We analyze the runtime of Thm. 4.5 in App. C.1.

## 5 CASE STUDIES IN THREE SETTINGS

We now compute the two quantities from §3, the post-hoc conditioning estimate and the interventional 'do'-estimate, in three case studies. We conduct two types of interventions using the binning semiring (§4.1). First, using **at-least-once** interventions, where we sample corpora such that the property occurs at least once in $N$ strings in the corpus of size $K$. This corresponds to having two bins with all probability mass in the second bin. We also consider a more **general binning** setting, where we see exactly $K$ occurrences in all strings; here we track occurrences in $N+1$ bins. In general, we train more models on the non-intervened data to compare against, as the chances of seeing a low number of occurrences are generally low for large corpora and machines. Details on the sampling of machines are given App. E. Details on the training configurations are given in App. F. All errors are standard errors. Since we wish to know how the frequency of a *targeted feature* impacts learnability, we derive a **decomposed KL divergence** (see App. D for details) between the trained model $p_{\boldsymbol{\theta}}$ and the PFSA, this serves as the M in the causal diagram given in §3.

**Decomposed KL example for states.** We give an example of the decomposed KL here, that for state interventions, transition and symbol interventions are given in App. D. Let $p_{\mathcal{A}}$ be the LM defined

---

[5]This corresponds to the case when $N$ occurrences are required. If some other number of occurrences $n$ is required, $\mathcal{A}'$ can simply be modified such that $(q_\iota, n)$ is the initial state.

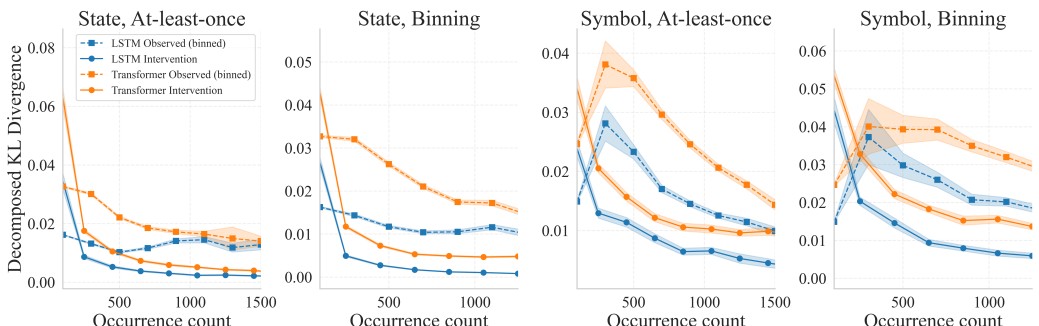

Figure 4: Monte Carlo estimates of the relation between learnability and the number of symbol occurrences over machines with 50 states and 10 symbols, for both states and symbols.

by a DPFSA $\mathcal{A}$ over states $Q$ and symbols $\Sigma$, and let $p_\theta$ be the trained LM. For a dataset property $P$, let $\pi(q) \stackrel{\text{def}}{=} \text{Pr}_{\boldsymbol{\sigma} \sim p_\mathcal{A}}\big[\delta(\boldsymbol{\sigma}) = q\big]$, and $\pi(q,a) \stackrel{\text{def}}{=} \pi(q)p_\mathcal{A}(a \mid q)$. Then the *state-wise* decomposition is

$$D_{\text{KL}}(p_\mathcal{A} \mid\mid p_\theta) = \sum_{q \in Q} \pi(q) \underbrace{\sum_{\substack{\boldsymbol{\sigma} \in \Sigma^* \\ \delta(\boldsymbol{\sigma}) = q}} \frac{p_\mathcal{A}(\boldsymbol{\sigma})}{\pi(q)} D_{\text{KL}}\big(p_\mathcal{A}(\cdot \mid q)\big\| p_\theta(\cdot \mid \boldsymbol{\sigma})\big)}_{\text{unweighted per-state contribution}}, \tag{14}$$

We calculate the divergence in Eq. (14) under intervention and observation to get the estimands in §3.

**Parity + Free Machine.** We sample weights over the topology shown in Fig. 1. We sample 200 different configurations of the machine, under intervention from 80 of those, and use all for observational data. For each machine and training configuration, we sample 500 samples, and train ten models. Results for the general binning case are shown in Fig. 2 and the at-least-once case in Fig. 6. As described in §3 we see a clear separation of trends when sampling causally for the accepting states. They appear more sample-demanding for the LSTM in this setup than the 'Odd' state, while the 'Free' state appears more complex for the Transformer.

**Aggregating Over Topologies.** We sample 1000 machines over the space of machines with 50 states, and 10 symbols. For 100 of those, we conduct interventions, and sample 10 weight settings for each machine. Here, we sample 2000 examples and train a single model per configuration. Results for symbol and state interventions are shown in Fig. 4. In both cases, there is a clear distinction between the learnability of the languages sampled on intervened on vs. observed data. In the former, we see an interesting phenomenon: not only is there a clear difference in the decomposed KL, but for lower occurrence counts, there is an inverse trend indicating complex structural confounders for machines that naturally produce a low number of occurrences.

**Varying Only Weights.** Finally, we sample a machine with 40 states and 10 symbols and a constant unweighted transition configuration. We sample 400 weight configurations for the observed data, and 10 configurations for the interventions using the at-least-once approach. We train 5 machines for each configuration using the LSTM and Transformer architectures. We visualize 8 random states in Fig. 5. We see, again, that there is a distinction between observation and correlation at all states (note the log-scale), although with some states less separated (9 and 1) than others. Notably, some states are more sample-demanding (complex) to learn than others, state 3 for both architectures and state 4 for the transformer. And some states are more so than others, such as state 3 and 6.

## 6   RELATED WORK

Much work has used formal languages to probe neural networks (Cleeremans et al., 1989; Jacobsson, 2005; Valvoda et al., 2022; Svete et al., 2024b; Borenstein et al., 2024, *inter alia*). A central empirical tool is synthetic data: SCAN-style setups examine compositional generalization (Lake & Baroni, 2018; Bastings et al., 2018; Ruis et al., 2020), and $k$-Dyck languages probe nested structure (Weiss et al., 2018; Suzgun et al., 2019; Bhattamishra et al., 2020; Hewitt et al., 2020). Recent work also

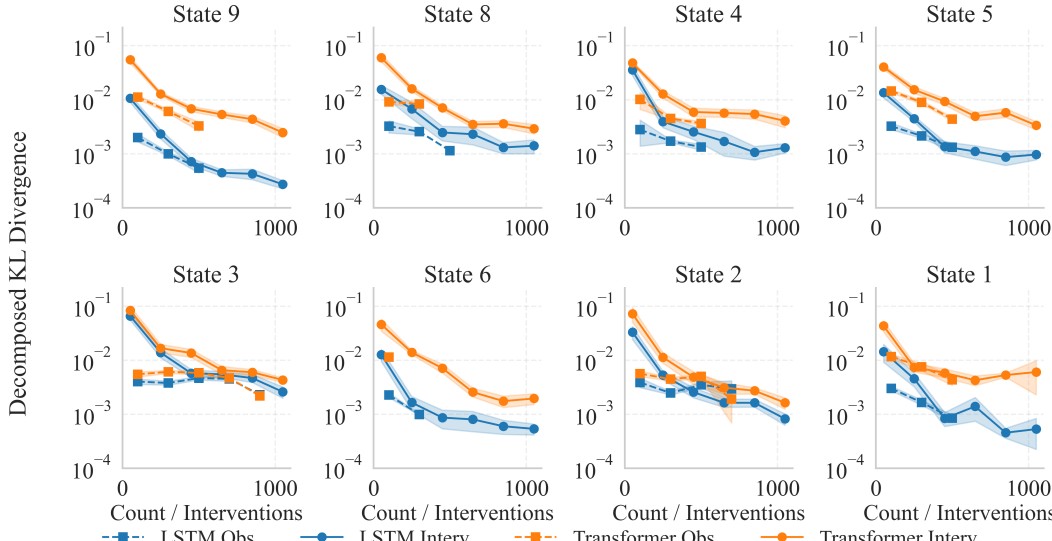

Figure 5: MCE of the Decomposed KL at randomly chosen states in a sampled automaton, with varying weight configurations. We see how observational results can lead to misleading assumptions.

analyzes inductive biases relative to the Chomsky hierarchy (Delétang et al., 2023; Butoi et al., 2024) and studies whole *classes* of languages rather than single datasets (Valvoda et al., 2022; Borenstein et al., 2024), continuing a line of grammatical inference research (Jacobsson, 2005). Complementing these empirical traditions, theory investigates the representational capacity and internal mechanisms of neural LMs and architectures, especially Transformers (Merrill, 2023; Strobl et al., 2023; Merrill, 2019; Merrill et al., 2020; Liu et al., 2023). While insightful, such analyses often rely on idealized assumptions and yield bounds that do not directly characterize practical learning behavior.

By contrast, there has been relatively little emphasis on *causal* approaches to LM behavior. Causal investigation with natural language is difficult—requiring complex taxonomies (Chen et al., 2024) or targeted neuron-level interventions (Vig et al., 2020; Finlayson et al., 2021). Our focus is *learnability under standard, fixed-size training regimes and natural data*, not strict expressivity. We use parity as a compact case study to illustrate our decomposition, controlled sampling, and causal approach; it also serves as a touchstone to situate prior art–from fixed-size impossibility (Hahn, 2020), through engineered solutions (Chiang & Cholak, 2022), and protocol changes such as chain-of-thought that add compute (Kim & Suzuki, 2025; Wen et al., 2024). Our work is orthogonal to these, and complements broader accounts of, e.g., Transformer capability (Allen-Zhu & Li, 2025).

## 7 CONCLUSION

We present a causal framework for assessing learnability in PFSA-induced language models. By replacing unconstrained sampling with controlled frequency interventions, we separate two drivers of observed difficulty—*frequency scarcity* and the learner's sample *complexity*. Across three case studies, interventional estimates often differ from correlational ones, illustrating the scarcity—complexity collision. The framework offers a way to diagnose whether difficulty reflects rarity in data or the learner–task pair complexity, and can inform dataset design and failure analysis in controlled settings.

**Limitiations** We focus on PFSA-induced settings where properties are exactly countable; applying the framework to natural language would require approximate labels for target phenomena. The sampler's cost grows with automaton size. We evaluate fixed-size learners under vanilla training. Our results concern the causal evaluation procedure as such, controlled sampling and evaluation, not a particular architecture; LSTM/Transformer are illustrative choices of $L$, and the framework extends to alternative learners (e.g., Mamba Gu & Dao (2024), CoT-augmented decoding Wei et al. (2022)) without modification.

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

## A  PRELIMINARIES

This section introduces the technical background introduced in the main paper.

### A.1  FORMAL LANGUAGE THEORY

Let $\mathbb{K}$ be a set, $\odot$ a binary operation, and $\mathbf{1} \in \mathbb{K}$. We say $(\mathbb{K}, \odot, \mathbf{1})$ is a **monoid** if

*(i)* $\mathbb{K}$ is closed under $\odot$,
*(ii)* $\odot$ is associative, and
*(iii)* $\mathbf{1}$ is the unit of $\odot$: $\forall a \in \mathbb{K} : \mathbf{1} \odot a = a \odot \mathbf{1} = a$;

We say that a monoid is **commutative** if $\forall a, b \in \mathbb{K}:  a \odot b = b \odot a$.

A **semiring** is a tuple $(\mathbb{K}, \oplus, \otimes, \mathbf{0}, \mathbf{1})$ where

*(i)* $(\mathbb{K}, \oplus, \mathbf{0})$ is a commutative monoid,
*(ii)* $(\mathbb{K}, \otimes, \mathbf{1})$ is a monoid,
*(iii)* multiplication left- and right-distributes over addition: $a \otimes (b \oplus c) = (a \otimes b) \oplus (a \otimes c)$ and $(b \oplus c) \otimes a = (b \otimes a) \oplus (c \otimes a)$, and
*(iv)* multiplication with $\mathbf{0}$ annihilates $\mathbb{K}$: $\mathbf{0} \otimes a = a \otimes \mathbf{0} = \mathbf{0}$.

A semiring admitting well-behaved infinite sums is called **complete**. In a complete semiring, we can define $a^* \stackrel{\text{def}}{=} \bigoplus_{i=0}^{\infty} a^i = \bigoplus_{i=0}^{\infty} \bigotimes_{j=1}^{i} a$. The **real semiring** is $(\mathbb{R}, +, \times, 0, 1)$ with the standard addition and multiplication. Adjoining $\infty$ to positive reals in the natural way gives a complete semiring with $a^* = \frac{1}{1-a}$ for $a < 1$ and $a^* = \infty$ otherwise.

A **weighted finite-state automaton** (WFSA) $\mathcal{A}$ over a semiring $(\mathbb{K}, \oplus, \otimes, \mathbf{0}, \mathbf{1})$ is a tuple $(Q, \Sigma, \delta, \lambda, \rho)$ where $Q$ is a finite set of states; $\Sigma$ is an alphabet; $\delta \colon Q \times \Sigma \times Q \to \mathbb{K}$ is a transition function, where we say that $\mathcal{A}$ has transition $p \xrightarrow{a/w} q$ with weight $w$ if $\delta(p, a, q) = w$; and $\lambda \colon Q \to \mathbb{K}$ and $\rho \colon Q \to \mathbb{K}$ are the initial and final weight functions, respectively.

A **path** $\boldsymbol{\pi}$ in $\mathcal{A}$ is a finite sequence of consecutive transitions $q_0 \xrightarrow{a_1/w_1} q_1 \cdots q_{N-1} \xrightarrow{a_N/w_N} q_N$. We write $|\boldsymbol{\pi}| = N$ for $\boldsymbol{\pi}$'s length. The **inner weight** of $\boldsymbol{\pi}$ is $\overline{w}(\boldsymbol{\pi}) \stackrel{\text{def}}{=} w_1 \otimes \cdots \otimes w_N$, its **weight** is $\boldsymbol{w}(\boldsymbol{\pi}) \stackrel{\text{def}}{=} \lambda(q_0) \otimes \overline{\boldsymbol{w}}(\boldsymbol{\pi}) \otimes \rho(q_N)$, and its **yield** is $a_1 \cdots a_N$. $\Pi(\mathcal{A})$ denotes the set of all paths in $\mathcal{A}$. We say that $\mathcal{A} = (Q, \Sigma, \delta, \lambda, \rho)$ is **deterministic** (a DPFSA) if, for every $p \in Q, \sigma \in \Sigma$, there is at most one $q \in Q$ such that $p \xrightarrow{a/w} q \in \delta$ with $w \neq \boldsymbol{0}$, and there is a single state $q_\iota$ with $\lambda(q_\iota) \neq \boldsymbol{0}$. In this case, we refer to $q_\iota$ as the **initial state**. Naturally, a deterministic WFSA can have at most one path with non-$\boldsymbol{0}$ path weight yielding a string $\boldsymbol{\sigma} \in \Sigma^*$ from the initial state $q_\iota$.

We say that a real-weighted WFSA with non-negative weights is **probabilistic** (a PFSA) if, for all states $q \in Q, \delta, \lambda$ and $\rho$ satisfy $\sum_{q \in Q} \lambda(q) = 1$, and $\rho(q) + \sum_{q \xrightarrow{\sigma/w} q' \in \delta} w = 1$. A PFSA induces a probability distribution over $\Sigma^*$, where the probability of a string $\boldsymbol{\sigma}$ is the sum of the weights of all paths yielding $\boldsymbol{\sigma}$ from the initial state $q_\iota$.

The **backward weight** $\boldsymbol{\beta}_{\mathcal{A}}(q)$ of a state $q \in Q$ is the sum of the weights of all paths starting at $q$: $\boldsymbol{\beta}_{\mathcal{A}}(q) \stackrel{\text{def}}{=} \bigoplus_{\boldsymbol{\pi} \in \Pi(\mathcal{A}), \iota(\boldsymbol{\pi})=q} \overline{w}(\boldsymbol{\pi}) \otimes \rho(\varphi(\boldsymbol{\pi}))$, where $\iota(\boldsymbol{\pi})$ and $\varphi(\boldsymbol{\pi})$ denote the origin and the destination states of $\boldsymbol{\pi}$, respectively. Finally, the **allsum** of $\mathcal{A}$ is $\boldsymbol{Z} \stackrel{\text{def}}{=} \sum_{\boldsymbol{\pi} \in \Pi(\mathcal{A})} \boldsymbol{w}(\boldsymbol{\pi}) = \boldsymbol{\beta}_{\mathcal{A}}(q_\iota)$.

## A.2 MARKOV KERNELS AND GRAPHICAL CAUSAL MODELS

**Notation.** We denote sets with uppercase caligraphic font, e.g., $\mathcal{X}$ and $\mathcal{Y}$, and $\sigma$-algebras in uppercase Fraktur e.g., $\mathfrak{X}$ and $\mathfrak{Y}$. We write $(\mathcal{X}, \mathfrak{X})$ for measurable spaces. We denote an element of a set $\mathcal{X}$ as a lowercase letter $x$. Random variables over $\mathcal{X}$ are typeset as uppercase, unitalicized X.

We model targeted data modifications as interventions in the graphical model in §3. In particular, §3 describes the process of training and evaluating an LM L from data D sampled from the ground-truth LM A with the following components: *(i)* $(\mathcal{A}, \mathfrak{A})$ is a probability space over some ground truth LM. We use A to denote a random variable over this space and $a$ for an element in the space. *(ii)* $(\mathcal{D}, \mathfrak{D})$ is a probability space over a corpus of size $K \in \mathbb{N}$. D is a random variable and $d$ a specific instantiation. *(iii)* $(\mathcal{L}, \mathfrak{L})$ is a probability space of all configurations of a given neural LM, where $\mathfrak{L}$ is a given $\sigma$-algebra. A random variable from the space is given by L and an element in $\mathcal{L}$ with $l$. *(iv)* $(\mathcal{M}, \mathfrak{M})$ is a probability space over the values that a comparison (measure of fit) between $a$ and $l$ can take. In §4, we focus on a specific specification of this model.

We want to define a causal model that captures the effect of intervening on the training data when training a neural LM. For the formal treatment of causal relationships in continuous spaces, we build on the framework of graphical causal models as presented in Peters et al. (2017).

**Notation.** We adopt the following notation: sets are denoted with uppercase caligraphic font, e.g., $\mathcal{X}$ and $\mathcal{Y}$, and elements of a set $\mathcal{X}$ as a lowercase letter $x$. Random variables over $\mathcal{X}$ are typeset as uppercase, unitalicized X.

Following Peters et al. (2017), we define a **graphical causal model** (GCM) as a triplet $\mathcal{G} = (\boldsymbol{V}, \boldsymbol{E}, \mathbf{K})$ where $\boldsymbol{V} = \{V_1, \ldots, V_N\}$ is a finite set of random variables over domains $\mathcal{V}_1, \ldots, \mathcal{V}_N$, $\boldsymbol{E} \subset \boldsymbol{V} \times \boldsymbol{V}$ is a set of edges forming a directed acyclic graph (DAG), and $\mathbf{K} = \{\kappa_1, \ldots, \kappa_N\}$ is a set of Markov kernels.

For each random variable $V_j \in \boldsymbol{V}$, the set $\text{PA}_j \subset \boldsymbol{V}$ denotes the **parent variables** of $V_j$ according to the DAG. The probabilistic behavior of the GCM is specified via a collection of Markov kernels, where each $\kappa_j$ describes the relation between a variable and all of its parents, including cases with multiple parents. The joint distribution over all variables is then induced by the factorization:

$$p(x_1, \ldots, x_N) = \prod_{j=1}^{N} p(x_j \mid x_{\text{PA}_j}) \tag{15}$$

Here, each conditional distribution $p(x_j \mid x_{\text{PA}_j})$ corresponds to the Markov kernel $\kappa_j$, which takes all parent variables as inputs.

This formulation allows us to handle intervention distributions in continuous spaces through covariate adjustment formulas as described in Peters et al. (2017, Section 6.6). In our work, we apply this framework specifically to model interventions on neural LM training data.

# B   THE BINNING SEMIRING

## B.1   MONOID SEMIALGEBRAS

**Definition B.1.** A **cyclic monoid** is a monoid with a single generator $c$ and an identity element. We denote by $C_{N,1}$ the monoid $C_{N,1} \stackrel{\text{def}}{=} \{c^0, c^1, \ldots, c^N\}$ where $c^0$ is the identity. The multiplication is defined for $0 \leqslant i, j \leqslant N$ as

$$c^i \cdot c^j \stackrel{\text{def}}{=} \begin{cases} c^j = c^{i+j} & \textbf{if } i + j < N \\ c^N & \textbf{otherwise} . \end{cases} \tag{16}$$

In particular, $c^N \cdot c^1 = c^N$.

**Definition B.2.** Given a finite monoid $M$ and a (semi)ring $(\mathbb{K}, \oplus, \otimes, \mathbf{0}, \mathbf{1})$, the **monoid semialgebra** $\mathbb{K}[M]$ consists of all formal finite sums

$$f = \sum_{m \in M} a_m m \tag{17}$$

where only finitely many $a_m \in \mathbb{K}$ are non-zero, with point-wise addition. The product in $\mathbb{K}[M]$ is given by a convolution:

$$\Big( \sum_{m \in M} a_m m \Big) \cdot \Big( \sum_{n \in M} b_n n \Big) = \sum_{m' \in M} \Big( \sum_{\substack{m, n \in M \\ mn = m'}} a_m b_n \Big) m' \tag{18}$$

Applying Definition B.2 to Definition B.1, we can see that element of $[\mathbb{K}]$ can be written uniquely as $\sum_{i=0}^{N} a_i c^i$ where $a_i \in \mathbb{K}$ and the convolution product is given by

$$\Big( \sum_{i=0}^{N} a_i c^i \Big) \cdot \Big( \sum_{i=0}^{N} b_i c^i \Big) = \sum_{i=0}^{N-1} \Big( \sum_{j=0}^{i} a_j b_{i-j} \Big) c^i + \Big( \sum_{\substack{i,j=0 \\ i+j \geqslant N}} a_i b_j \Big) c^N \tag{19}$$

## B.2   A CLOSED-FORM SOLUTION FOR THE KLEENE STAR

Computing backward weights in a general PFSA with cycles requires summation over infinitely many paths. This can be done efficiently with *asteration*—taking the Kleene closure of the set of paths. This section develops the Kleene star operator for binning automata over PFSAs and records when the resulting solution is unique.

**Lemma B.3** (Arden's rule in complete semirings). *Let* $(\mathbb{K}, \oplus, \otimes, \mathbf{0}, \mathbf{1})$ *be a complete semiring. For all* $a, b \in \mathbb{K}$, *the equation*

$$x = (a \otimes x) \oplus b \tag{20}$$

*has* $x = a^* \otimes b$ *as a solution.*

*Proof.* By Kuich (1997, Thm. 2.2(ii)), $a^* = \mathbf{1} \oplus (a \otimes a^*)$. Therefore

$$\begin{aligned} (a \otimes (a^* \otimes b)) \oplus b &= (a \otimes a^*) \otimes b \oplus b \\ &= ((a \otimes a^*) \oplus \mathbf{1}) \otimes b \\ &= a^* \otimes b. \end{aligned} \tag{21}$$

∎

**Proposition B.4** (Coefficientwise Kleene equations in the binning semiring). *Let* $[\mathbb{K}]$ *be the* $N$*th-order binning semiring and* $\boldsymbol{v} \in [\mathbb{K}]$. *Write* $\boldsymbol{v}^* \stackrel{\text{def}}{=} \bigoplus_{i=0}^{\infty} \boldsymbol{v}^{\otimes i}$. *For* $0 \leqslant i < N$,

$$(\boldsymbol{v}^*)_i = v_0^* \otimes \Big( (\mathbf{1})_i \oplus \bigoplus_{j=1}^{i} v_j \otimes (\boldsymbol{v}^*)_{i-j} \Big). \tag{22}$$

*For the overflow bin,*

$$(\boldsymbol{v}^*)_N = \Big( \bigoplus_{i=0}^{N} v_i \Big)^* \otimes \Big( \bigoplus_{\substack{i,j=0 \\ j \neq N \\ i+j \geqslant N}}^{N} v_i \otimes (\boldsymbol{v}^*)_j \Big). \tag{23}$$

*Proof.* For $0 \leqslant i < N$,

$$
\begin{aligned}
(\boldsymbol{v}^*)_i = \left( \bigoplus_{k=0}^{\infty} \boldsymbol{v}^{\otimes k} \right)_i &= \textcircled{1}_i \oplus (\boldsymbol{v} \otimes \boldsymbol{v}^*)_i \\
&= \textcircled{1}_i \oplus \bigoplus_{j=0}^{i} v_j \otimes (\boldsymbol{v}^*)_{i-j} \\
&= v_0 \otimes (\boldsymbol{v}^*)_i \oplus \left( \textcircled{1}_i \oplus \bigoplus_{j=1}^{i} v_j \otimes (\boldsymbol{v}^*)_{i-j} \right).
\end{aligned}
\tag{24}
$$

Apply Lemma B.3 with $a = v_0$ to obtain (22). For $i = N$,

$$
\begin{aligned}
(\boldsymbol{v}^*)_N = \textcircled{1}_N \oplus (\boldsymbol{v} \otimes \boldsymbol{v}^*)_N &= \textcircled{1}_N \oplus \bigoplus_{\substack{i,j=0 \\ i+j \geqslant N}}^{N} v_i \otimes (\boldsymbol{v}^*)_j \\
&= \left( \bigoplus_{i=0}^{N} v_i \right) \otimes (\boldsymbol{v}^*)_N \oplus \bigoplus_{\substack{i,j=0 \\ j \neq N, i+j \geqslant N}}^{N} v_i \otimes (\boldsymbol{v}^*)_j,
\end{aligned}
\tag{25}
$$

and Lemma B.3 with $a = \bigoplus_{i=0}^{N} v_i$ yields (23). ∎

**Lemma B.5** (Uniqueness in the binning semiring). *Let $(\mathbb{R}_{\geqslant 0}, +, \cdot, 0, 1)$ be the non-negative real semiring, and let $[\mathbb{K}]$ be the binning semiring of order $N$ over $\mathbb{R}_{\geqslant 0}$. For any $v, w \in [\mathbb{K}]$ with $v$ arising from a PFSA binning automaton and $v_0 < 1$, the equation*

$$
x = (v \otimes x) \oplus w
\tag{26}
$$

*has a unique solution.*

*Proof.* For $0 \leqslant i < N$,

$$
x_i = v_0 \otimes x_i \oplus \bigoplus_{j=1}^{i} v_j \otimes x_{i-j} \oplus w_i.
\tag{27}
$$

Working in $\mathbb{R}_{>0}$ and using $v_0 < 1$,

$$
x_i = \frac{1}{1 - v_0} \cdot \left( \bigoplus_{j=1}^{i} v_j \otimes x_{i-j} \oplus w_i \right),
\tag{28}
$$

which uniquely determines $x_i$ by strong induction and preserves non-negativity. For $i = N$, collect overflow terms to get

$$
(1 - v_0) \cdot x_N = \bigoplus_{j=1}^{N} v_j \otimes x_{N-j} \oplus \bigoplus_{\substack{i+j \geqslant N \\ i,j > 0}} v_i \otimes x_j \oplus w_N,
\tag{29}
$$

and the same argument applies. ∎

**Corollary B.6** (Uniqueness of the closed form in the PFSA/binning setting). *Under the conditions of Proposition B.4 and Lemma B.5, the solution given by (22)–(23) is unique.*

# C  PROOFS

**Theorem 4.1** (Path Weight Interpretation). *For $\boldsymbol{\pi} \in \Pi(\mathcal{A}_\phi)$ and $i \in \{0, \dots, N-1\}$, we have:*

$$
\overline{\boldsymbol{w}}_\phi(\boldsymbol{\pi})_i = \begin{cases} \overline{\boldsymbol{w}}(\boldsymbol{\pi}) & \textbf{if } i < N \text{ and } |\boldsymbol{\pi}|_\phi = i \text{ or } i = N \text{ and } |\boldsymbol{\pi}|_\phi \geqslant N \\ 0 & \textbf{otherwise} \end{cases}
\tag{8}
$$

*where $\overline{\boldsymbol{w}}(\boldsymbol{\pi})$ is the inner path weight in $\mathcal{A}$.*

*Proof.* We proceed by induction on $|\boldsymbol{\pi}|$.

If $|\boldsymbol{\pi}| = 1$, then $\overline{\boldsymbol{w}}_\phi(\boldsymbol{\pi}) = \mathcal{L}_{\phi_\delta}(q_0 \xrightarrow{a_1/w_1} q_1)$, so the claim follows from the definition of $\mathcal{L}_{\phi_\delta}$.

Suppose now that Eq. (8) holds for all paths $\boldsymbol{\pi}$ with $|\boldsymbol{\pi}| < m$. Let $\boldsymbol{\pi} = q_0 \xrightarrow{a_1/\boldsymbol{w}_1} q_1 \cdots \xrightarrow{a_m/\boldsymbol{w}_m} q_m$ be a path of length $m$, and $\boldsymbol{\pi}' = q_0 \xrightarrow{a_1/\boldsymbol{w}_1} q_1 \cdots \xrightarrow{a_{m-1}/\boldsymbol{w}_{m-1}} q_{m-1}$ be the sub-path of the first

$m - 1$ transitions. We compute for $i \in \{0, \ldots, N - 1\}$

$$\overline{w}(\pi)_i = (\overline{w}(\pi') \circledcirc w_m)_i \tag{30a}$$

$$= \sum_{j=0}^{i} \overline{w}(\pi')_j \otimes (w_m)_{i-j} \tag{30b}$$

$$= \overline{w}(\pi')_{|\pi'|_\phi} \otimes (w_m)_{i-|\pi'|_\phi} \qquad \text{(Inductive hypothesis, 30c)}$$

$$= \begin{cases} \overline{w}(\pi') \otimes w_m = \overline{w}(\pi) & \text{if } i = |\pi|_\phi + \phi_\delta(q_{m-1} \xrightarrow{a_m/w_m} q_m) \\ \mathbf{0} & \text{otherwise} \end{cases} . \tag{30d}$$

This completes the proof for coefficients $i < N$. For the remaining case, we derive:

$$\overline{w}(\pi)_N = (\overline{w}(\pi') \circledcirc w_m)_N \tag{31a}$$

$$= \sum_{\substack{i,j=0 \\ i+j \geqslant N}}^{N} \overline{w}(\pi')_i \otimes (w_m)_j \tag{31b}$$

$$= \begin{cases} \overline{w}(\pi')_{|\pi'|_\phi} \otimes (w_m)_{\phi_\delta(q_{m-1} \xrightarrow{a_m/w_m} q_m)} & \text{if } |\pi'|_\phi + \phi_\delta(q_{m-1} \xrightarrow{a_m/w_m} q_m) \geqslant N \\ \mathbf{0} & \text{otherwise} \end{cases}$$

$$\text{(Inductive hypothesis, 31c)}$$

$$= \begin{cases} \overline{w}(\pi') \otimes w_m = \overline{w}(\pi) & \text{if } |\pi'|_\phi + \phi_\delta(q_{m-1} \xrightarrow{a_m/w_m} q_m) \geqslant N \\ \mathbf{0} & \text{otherwise} \end{cases} . \tag{31d}$$

$$\blacksquare$$

**Theorem 4.2** (Backward Weight Interpretation). *For $q \in Q$, we have:*

$$\boldsymbol{\beta}_\phi(q)_i = \begin{cases} \mathbb{P}(|\boldsymbol{\Pi}(q)|_\phi = i) & \textit{if } i < N \\ \mathbb{P}(|\boldsymbol{\Pi}(q)|_\phi \geqslant N) & \textit{otherwise} \end{cases} \tag{10}$$

*Proof.* For any $i \in \{0, \ldots, N - 1\}$, we have:

$$p(|\boldsymbol{\Pi}(q)|_\phi = i) = \sum_{\substack{\pi \in \Pi(\mathcal{A}) \\ |\pi|_\phi = i}} p(\pi) \tag{32a}$$

$$= \sum_{\substack{\pi \in \Pi(\mathcal{A}) \\ |\pi|_\phi = i}} w(\pi) \qquad (\mathcal{A} \text{ is a PFSA, 32b})$$

$$= \sum_{\substack{\pi \in \Pi(\mathcal{A}_\phi) \\ |\pi|_\phi = i}} w_\phi(\pi)_{|\pi|_\phi} \qquad (\text{Thm. 4.1 , 32c})$$

$$= \sum_{\pi \in \Pi(\mathcal{A}_\phi)} w_\phi(\pi)_i \qquad (\text{Thm. 4.1 , 32d})$$

$$= \boldsymbol{\beta}_\phi(q)_i. \qquad (\text{Definition of } \boldsymbol{\beta}_\phi, \text{ 32e})$$

Derivation for coefficients $i = N$ is analogous. $\blacksquare$

**Theorem 4.3** (Counting in a Set). *Denoting with $\boldsymbol{Z} \overset{\text{def}}{=} \boldsymbol{\beta}_\phi(q_\iota) = \sum_{\pi \in \Pi(\mathcal{A}_\phi)} w_\phi(\pi)$ the sum of all path weights in the binning automaton $\mathcal{A}_\phi$, we have:*

$$(\boldsymbol{Z}^{\otimes K})_i = \begin{cases} \mathbb{P}(|\mathcal{P}|_\phi = i) & \textit{if } i < N \\ \mathbb{P}(|\mathcal{P}|_\phi \geqslant N) & \textit{otherwise} \end{cases} \tag{11}$$

*Proof.* Follows immediately from Thm. 4.2 and the definition of $\boldsymbol{Z}$. $\blacksquare$

**Theorem 4.4** (Sampling Path Event Counts (1)). *We have that*

$$\mathbb{P}(|\boldsymbol{\pi}_k|_\phi = i \mid |\mathcal{P}|_\phi = N, |\{\boldsymbol{\pi}_j\}_{j<k}|_\phi = m) = \frac{(\boldsymbol{Z}^{\otimes K-k-1})_{N-m-i}}{(\boldsymbol{Z}^{\otimes K-k})_{N-m}} \boldsymbol{Z}_i \qquad (12)$$

*Proof.* To derive Eq. (12), we have that

$$\mathbb{P}(|\boldsymbol{\pi}_k|_\phi = i \mid |\mathcal{P}|_\phi = N, |\{\boldsymbol{\pi}_j\}_{j<k}|_\phi = m) = \frac{\mathbb{P}(|\boldsymbol{\pi}_k|_\phi = i, |\mathcal{P}|_\phi = N, |\{\boldsymbol{\pi}_j\}_{j<k}|_\phi = m)}{\mathbb{P}(|\mathcal{P}|_\phi = N, |\{\boldsymbol{\pi}_j\}_{j<k}|_\phi = m)}. \qquad (33)$$

With this, we derive

$$\mathbb{P}(|\boldsymbol{\pi}_k|_\phi = i, |\mathcal{P}|_\phi = N, |\{\boldsymbol{\pi}_j\}_{j<k}|_\phi = m) \qquad (34a)$$

$$= \mathbb{P}(|\boldsymbol{\pi}_k|_\phi = i, |\{\boldsymbol{\pi}_j\}_{j>k+1}|_\phi = N - m - i, |\{\boldsymbol{\pi}_j\}_{j<k}|_\phi = m) \qquad (34b)$$

$$= \mathbb{P}(|\boldsymbol{\pi}_k|_\phi = i)\mathbb{P}(|\{\boldsymbol{\pi}_j\}_{j>k+1}|_\phi = N - m - i)\mathbb{P}(|\{\boldsymbol{\pi}_j\}_{j<k}|_\phi = m) \qquad \text{(Indepencence of samples, 34c)}$$

$$= \boldsymbol{Z}_i(\boldsymbol{Z}^{K-k-1})_{N-m-i}(\boldsymbol{Z}^{k-1})_m \qquad \text{(Thm. 4.3, 34d)}$$

and

$$\mathbb{P}(|\mathcal{P}|_\phi = N, |\{\boldsymbol{\pi}_j\}_{j<k}|_\phi = m) \qquad (35a)$$

$$= \mathbb{P}(|\{\boldsymbol{\pi}_j\}_{j\geq k}|_\phi = N - m, |\{\boldsymbol{\pi}_j\}_{j<k}|_\phi = m) \qquad (35b)$$

$$= \mathbb{P}(|\{\boldsymbol{\pi}_j\}_{j\geq k}|_\phi = N - m)\mathbb{P}(|\{\boldsymbol{\pi}_j\}_{j<k}|_\phi = m) \qquad \text{(Indepencence of samples, 35c)}$$

$$= (\boldsymbol{Z}^{K-k})_{N-m}(\boldsymbol{Z}^{k-1})_m \qquad \text{(Thm. 4.3, 35d)}$$

∎

We now consider sampling under the constraint $|\boldsymbol{\Pi}(q_\iota)|_\phi \geq n$. For that, define $\boldsymbol{Z}'(q) \stackrel{\text{def}}{=} \boldsymbol{M}\boldsymbol{Z}(q)$, where $\boldsymbol{M}$ is the upper-triangular matrix of ones. The interpretation of $\boldsymbol{Z}(q)$ is as follows: Whereas $\boldsymbol{Z}(q)_i$ contains the mass of strings length $i$ (or at least $N$ for $i = N$), $\boldsymbol{Z}'(q)_i$ adds up the mass of all strings of length at least $i$ or longer (including those summarized in the last entry). Thus, $\boldsymbol{Z}'(q)_i$ is the probability of seeing at least $i$ occurrences of $\phi$ in a random path starting in $q$. Further, define $\boldsymbol{Z}' \stackrel{\text{def}}{=} \boldsymbol{Z}'(q_\iota)$.

**Theorem C.1** (Sampling Path Event Counts (2)). *We have that*

$$\mathbb{P}(|\boldsymbol{\pi}_k|_\phi \geq i \mid |\mathcal{P}|_\phi \geq N, |\{\boldsymbol{\pi}_j\}_{j<k}|_\phi = m) \qquad (36)$$

$$= \frac{\left(\boldsymbol{Z}'_i(\boldsymbol{Z}'^{K-k-1})_{N-m} + \sum_{j=1}^{N-m}(\boldsymbol{Z}')_{i+j}(\boldsymbol{Z}^{K-k-1})_{N-m-j}\right)}{(\boldsymbol{Z}'^{K-k})_{N-m}}$$

*Proof.* To derive Eq. (36), we have that

$$\mathbb{P}(|\boldsymbol{\pi}_k|_\phi \geq i \mid |\mathcal{P}|_\phi \geq N, |\{\boldsymbol{\pi}_j\}_{j<k}|_\phi = m) = \frac{\mathbb{P}(|\boldsymbol{\pi}_k|_\phi \geq i, |\mathcal{P}|_\phi \geq N, |\{\boldsymbol{\pi}_j\}_{j<k}|_\phi = m)}{\mathbb{P}(|\mathcal{P}|_\phi \geq N, |\{\boldsymbol{\pi}_j\}_{j<k}|_\phi = m)}. \qquad (37)$$

With this, we derive

$$\mathbb{P}(|\boldsymbol{\pi}_k|_\phi \geq i, |\mathcal{P}|_\phi \geq N, |\{\boldsymbol{\pi}_j\}_{j<k}|_\phi = m) \qquad (38a)$$

$$= \mathbb{P}(|\boldsymbol{\pi}_k|_\phi \geq i, |\{\boldsymbol{\pi}_j\}_{j\geq k}|_\phi \geq N - m, |\{\boldsymbol{\pi}_j\}_{j<k}|_\phi = m) \qquad (38b)$$

$$= \mathbb{P}(|\boldsymbol{\pi}_k|_\phi \geq i, |\{\boldsymbol{\pi}_j\}_{j>k}|_\phi \geq N - m, |\{\boldsymbol{\pi}_j\}_{j<k}|_\phi = m)$$

$$+ \sum_{j=1}^{i} \mathbb{P}(|\boldsymbol{\pi}_k|_\phi \geq i + j, |\{\boldsymbol{\pi}_j\}_{j>k}|_\phi = N - m - j, |\{\boldsymbol{\pi}_j\}_{j<k}|_\phi = m) \qquad (38c)$$

$$= \boldsymbol{Z}'_i(\boldsymbol{Z}'^{K-k-1})_{N-m}(\boldsymbol{Z}^{k-1})_m + \sum_{j=1}^{N-m} \boldsymbol{Z}'_{i+j}(\boldsymbol{Z}^{K-k-1})_{N-m-j}(\boldsymbol{Z}^{k-1})_m \qquad (38d)$$

$$= \left(\boldsymbol{Z}'_i(\boldsymbol{Z}'^{K-k-1})_{N-m} + \sum_{j=1}^{N-m} \boldsymbol{Z}'_{i+j}(\boldsymbol{Z}^{K-k-1})_{N-m-j}\right)(\boldsymbol{Z}^{k-1})_m \qquad (38e)$$

and

$$\mathbb{P}(|\mathcal{P}|_\phi \geqslant N, |\{\boldsymbol{\pi}_j\}_{j<k}|_\phi = m) \tag{39a}$$

$$= \mathbb{P}(|\{\boldsymbol{\pi}_j\}_{j\geqslant k}|_\phi \geqslant N - m)\mathbb{P}(|\{\boldsymbol{\pi}_j\}_{j<k}|_\phi = m) \tag{39b}$$

$$= (\boldsymbol{Z}^{K-k})_{N-m}(\boldsymbol{Z}^{k-1})_m \qquad\qquad (\text{Thm. 4.3, 39c})$$

$$\blacksquare$$

**Theorem C.2** (Sampling Path Event Counts (3)). *Let $\boldsymbol{Z}$ be the binning pathsum of order $N = 1$. Then*

$$\mathbb{P}(|\boldsymbol{\Pi}_k(q_\iota)|_\phi \geqslant 1 \mid |\mathcal{P}|_{\mathbb{1}\{\phi\}} = M, |\{\boldsymbol{\pi}_j\}_{j<k}|_{\mathbb{1}\{\phi\}} = m) = \frac{\binom{K-k-1}{M-m-1}}{\binom{K-k}{M-m}}. \tag{40}$$

*Proof.* We have

$$\mathbb{P}(|\boldsymbol{\Pi}_k(q_\iota)|_\phi \geqslant 1 \mid |\mathcal{P}|_{\mathbb{1}\{\phi\}} = M, |\{\boldsymbol{\pi}_j\}_{j<k}|_{\mathbb{1}\{\phi\}} = m) \tag{41a}$$

$$= \frac{\mathbb{P}(|\boldsymbol{\Pi}_k(q_\iota)|_\phi \geqslant 1, |\mathcal{P}|_{\mathbb{1}\{\phi\}} = M, |\{\boldsymbol{\pi}_j\}_{j<k}|_{\mathbb{1}\{\phi\}} = m)}{\mathbb{P}(|\mathcal{P}|_{\mathbb{1}\{\phi\}} = M, |\{\boldsymbol{\pi}_j\}_{j<k}|_{\mathbb{1}\{\phi\}} = m)}. \tag{41b}$$

Since the sampling is conditionally independent given the constraint, we have

$$\mathbb{P}(|\boldsymbol{\Pi}_k(q_\iota)|_\phi \geqslant 1, |\mathcal{P}|_{\mathbb{1}\{\phi\}} = M, |\{\boldsymbol{\pi}_j\}_{j<k}|_{\mathbb{1}\{\phi\}} = m) \tag{42a}$$

$$= \mathbb{P}(|\boldsymbol{\Pi}_k(q_\iota)|_\phi \geqslant 1, |\{\boldsymbol{\pi}_j\}_{j>k}|_{\mathbb{1}\{\phi\}} = M - m - 1, |\{\boldsymbol{\pi}_j\}_{j<k}|_{\mathbb{1}\{\phi\}} = m) \tag{42b}$$

$$= \boldsymbol{Z}_1 \cdot \mathbb{P}(|\{\boldsymbol{\Pi}_j\}_{j>k}|_{\mathbb{1}\{\phi\}} = M - m - 1) \cdot \mathbb{P}(|\{\boldsymbol{\pi}_j\}_{j<k}|_{\mathbb{1}\{\phi\}} = m) \qquad (\text{Thm. 4.2, 42c})$$

$$= \boldsymbol{Z}_1 \cdot \binom{K-k-1}{M-m-1}(\boldsymbol{Z}_1)^{M-m-1}(\boldsymbol{Z}_0)^{K-k-1-(M-m-1)} \cdot \mathbb{P}(|\{\boldsymbol{\pi}_j\}_{j<k}|_{\mathbb{1}\{\phi\}} = m) \tag{42d}$$

$$= \binom{K-k-1}{M-m-1}(\boldsymbol{Z}_1)^{M-m}(\boldsymbol{Z}_0)^{K-k-(M-m)} \cdot \mathbb{P}(|\{\boldsymbol{\pi}_j\}_{j<k}|_{\mathbb{1}\{\phi\}} = m). \tag{42e}$$

We also have

$$\mathbb{P}(|\mathcal{P}|_{\mathbb{1}\{\phi\}} = M, |\{\boldsymbol{\pi}_j\}_{j<k}|_{\mathbb{1}\{\phi\}} = m) \tag{43a}$$

$$= \mathbb{P}(|\{\boldsymbol{\pi}_j\}_{j\geqslant k}|_{\mathbb{1}\{\phi\}} = M - m, |\{\boldsymbol{\pi}_j\}_{j<k}|_{\mathbb{1}\{\phi\}} = m) \tag{43b}$$

$$= \mathbb{P}(|\{\boldsymbol{\Pi}_j\}_{j\geqslant k}|_{\mathbb{1}\{\phi\}} = M - m) \cdot \mathbb{P}(|\{\boldsymbol{\pi}_j\}_{j<k}|_{\mathbb{1}\{\phi\}} = m) \qquad (\text{Thm. 4.2, 43c})$$

$$= \binom{K-k}{M-m}(\boldsymbol{Z}_1)^{M-m}(\boldsymbol{Z}_0)^{K-k-(M-m)} \cdot \mathbb{P}(|\{\boldsymbol{\pi}_j\}_{j<k}|_{\mathbb{1}\{\phi\}} = m). \tag{43d}$$

Thus

$$\mathbb{P}(|\boldsymbol{\Pi}_k(q_\iota)|_\phi \geqslant 1 \mid |\mathcal{P}|_{\mathbb{1}\{\phi\}} = M, |\{\boldsymbol{\pi}_j\}_{j<k}|_{\mathbb{1}\{\phi\}} = m) \tag{44a}$$

$$= \frac{\mathbb{P}(|\boldsymbol{\Pi}_k(q_\iota)|_\phi \geqslant 1, |\mathcal{P}|_{\mathbb{1}\{\phi\}} = M, |\{\boldsymbol{\pi}_j\}_{j<k}|_{\mathbb{1}\{\phi\}} = m)}{\mathbb{P}(|\mathcal{P}|_{\mathbb{1}\{\phi\}} = M, |\{\boldsymbol{\pi}_j\}_{j<k}|_{\mathbb{1}\{\phi\}} = m)} \tag{44b}$$

$$= \frac{\binom{K-k-1}{M-m-1}(\boldsymbol{Z}_1)^{M-m}(\boldsymbol{Z}_0)^{K-k-(M-m)}}{\binom{K-k}{M-m}(\boldsymbol{Z}_1)^{M-m}(\boldsymbol{Z}_0)^{K-k-(M-m)}} \tag{44c}$$

$$= \frac{\binom{K-k-1}{M-m-1}}{\binom{K-k}{M-m}}. \tag{44d}$$

$$\blacksquare$$

**Theorem 4.5** (Constrained String Sampling (1)). *We have that*

$$\mathbb{P}(\boldsymbol{\Pi}(q_\iota) = \boldsymbol{\pi} \mid |\boldsymbol{\Pi}(q_\iota)|_\phi = N - 1) = \mathbb{P}(\boldsymbol{\Pi}'(q_\iota) = \boldsymbol{\pi}). \tag{13}$$

*Proof.* We first show that the weights sum to 1 for any $q \in Q$ and $i \in \{0, \ldots, N-1\}$. Note that by definition, we have the following equalities

$$\delta'((q,i), \mathsf{a}, (q',i)) \overset{\text{def}}{=} \frac{1}{\boldsymbol{\beta}_\phi(q)_{N-i}} \cdot \left(\mathcal{L}_{\phi_\delta}(q \xrightarrow{a/w} q') \otimes \boldsymbol{\beta}_\phi(q')\right)_{N-i} \tag{45a}$$

$$= \frac{1}{\boldsymbol{\beta}_\phi(q)_{N-i}} \cdot w \cdot \left(\boldsymbol{\beta}_\phi(q')\right)_{N-i-\phi_\delta(q \xrightarrow{a/w} q')} \tag{45b}$$

$$\rho'(q,i) \overset{\text{def}}{=} \frac{1}{\boldsymbol{\beta}_\phi(q)_{N-i}} \cdot \mathcal{L}_\rho(q)_{N-i} \tag{45c}$$

$$= \frac{1}{\boldsymbol{\beta}_\phi(q)_{N-i}} \cdot \mathbb{1}\{N-i = \phi_\rho(q)\}\rho(q) \tag{45d}$$

From the equality

$$\boldsymbol{\beta}(q) = \bigoplus_{q \xrightarrow{a/w} q'} w \otimes \boldsymbol{\beta}(q') + \rho(q) \tag{46}$$

in a general WFSA, we have that[6]

$$\left(\boldsymbol{\beta}_\phi(q)\right)_i = \bigoplus_{q \xrightarrow{a/\boldsymbol{w}} q'} \left(\boldsymbol{w} \otimes \boldsymbol{\beta}_\phi(q')\right)_i + \left(\rho_{\mathcal{A}_\phi}(q)\right)_i \tag{47}$$

$$= \bigoplus_{q \xrightarrow{a/w} q'} \left(\mathcal{L}_{\phi_\delta}(q \xrightarrow{a/w} q') \otimes \boldsymbol{\beta}_\phi(q')\right)_i + \left(\rho_{\mathcal{A}_\phi}(q)\right)_i \tag{48}$$

$$= \bigoplus_{q \xrightarrow{a/w} q'} w \cdot \boldsymbol{\beta}_\phi(q')_{i-\phi_\delta(q \xrightarrow{a/w} q')} + \left(\rho_{\mathcal{A}_\phi}(q)\right)_i \tag{49}$$

$$= \bigoplus_{q \xrightarrow{a/w} q'} w \cdot \boldsymbol{\beta}_\phi(q')_{i-\phi_\delta(q \xrightarrow{a/w} q')} + \mathbb{1}\{i = \phi_\rho(q)\}\rho(q) \tag{50}$$

---

[6]For conciseness, we assume $\boldsymbol{v}_m = 0$ for $m < 0$ and $m > N$.

We then have for $\boldsymbol{\pi} = (q_\iota, N) \xrightarrow{a_1/w_1} \cdots \xrightarrow{a_m/w_m} (q_m, N')$ that

$$\mathbb{P}(\boldsymbol{\Pi}'(q_\iota) = \boldsymbol{\pi}) = \big(\prod_{t=1}^{m} w_t\big)\rho'(q_m, N') \tag{51a}$$

$$= \prod_{t=1}^{m} \frac{1}{\big(\boldsymbol{\beta}_\phi(q_{t-1})\big)_{N-1-|\boldsymbol{\pi}_{<t}|_\phi}} \cdot \big(\mathcal{L}_{\phi\delta}(q_{t-1} \xrightarrow{a_t/w_t} q_t) \otimes \boldsymbol{\beta}_\phi(q_t)\big)_{N-1-|\boldsymbol{\pi}_{<t}|_\phi}$$
$$\cdot \frac{1}{\big(\boldsymbol{\beta}_\phi(q_m)\big)_{N-1-|\boldsymbol{\pi}_{<m}|_\phi}} \cdot \big(\mathcal{L}_{\phi\delta}(q_m)\big)_{N-1-|\boldsymbol{\pi}_{<m}|_\phi} \tag{51b}$$

$$= \prod_{t=1}^{m} \frac{1}{\big(\boldsymbol{\beta}_\phi(q_{t-1})\big)_{N-1-|\boldsymbol{\pi}_{<t}|_\phi}} \cdot w_t \cdot \big(\boldsymbol{\beta}_\phi(q_t)\big)_{N-1-|\boldsymbol{\pi}_{<t}|-\phi_\delta(q_{t-1} \xrightarrow{a_t/w_t} q_t)}$$
$$\cdot \frac{1}{\big(\boldsymbol{\beta}_\phi(q_m)\big)_{N-1-|\boldsymbol{\pi}_{<m}|_\phi}} \cdot \big(\mathcal{L}_{\phi\delta}(q_m)\big)_{N-1-|\boldsymbol{\pi}_{<m}|_\phi} \tag{51c}$$

$$= \prod_{t=1}^{m} \frac{1}{\big(\boldsymbol{\beta}_\phi(q_{t-1})\big)_{N-1-|\boldsymbol{\pi}_{<t}|_\phi}} \cdot w_t \cdot \big(\boldsymbol{\beta}_\phi(q_t)\big)_{N-1-|\boldsymbol{\pi}_{\leqslant t}|}$$
$$\cdot \frac{1}{\big(\boldsymbol{\beta}_\phi(q_m)\big)_{N-1-|\boldsymbol{\pi}_{<m}|_\phi}} \cdot \big(\mathcal{L}_{\phi\delta}(q_m)\big)_{N-1-|\boldsymbol{\pi}_{<m}|_\phi} \tag{51d}$$

$$= \frac{1}{\big(\boldsymbol{\beta}_\phi(q_\iota)\big)_N} \cdot \prod_{t=1}^{m} w_t \cdot \big(\mathcal{L}_{\phi\delta}(q_m)\big)_{N-1-|\boldsymbol{\pi}_{<m}|_\phi} \qquad \text{(Telescoping product, 51e)}$$

$$= \frac{1}{\big(\boldsymbol{\beta}_\phi(q_\iota)\big)_N} \cdot \prod_{t=1}^{m} w_t \cdot \mathbb{1}\{N - 1 - |\boldsymbol{\pi}_{<m}|_\phi = \phi_\rho(q)\} \cdot \rho(q) \tag{51f}$$

$$= \frac{1}{\big(\boldsymbol{\beta}_\phi(q_\iota)\big)_N} \cdot \prod_{t=1}^{m} w_t \cdot \rho(q) \cdot \mathbb{1}\{N - 1 - |\boldsymbol{\pi}_{<m}|_\phi = \phi_\rho(q)\} \tag{51g}$$

$$= \frac{1}{\big(\boldsymbol{\beta}_\phi(q_\iota)\big)_N} \cdot \mathbb{P}(\boldsymbol{\Pi}(q_\iota) = \boldsymbol{\pi}) \cdot \mathbb{1}\{N - 1 - |\boldsymbol{\pi}_{<m}|_\phi = \phi_\rho(q)\} \tag{51h}$$

$$= \frac{1}{\big(\boldsymbol{\beta}_\phi(q_\iota)\big)_N} \cdot \mathbb{P}(\boldsymbol{\Pi}(q_\iota) = \boldsymbol{\pi}, |\boldsymbol{\pi}|_\phi = N) \tag{51i}$$

$$= \frac{1}{\mathbb{P}(|\boldsymbol{\Pi}(q_\iota)|_\phi = N)} \cdot \mathbb{P}(\boldsymbol{\Pi}(q_\iota) = \boldsymbol{\pi}, |\boldsymbol{\pi}|_\phi = N) \tag{51j}$$

$$= \mathbb{P}(\boldsymbol{\Pi}(q_\iota) = \boldsymbol{\pi} \mid |\boldsymbol{\Pi}(q_\iota)|_\phi = N). \tag{51k}$$

$$\blacksquare$$

Let $M$ be the upper-triangular matrix of ones and $\boldsymbol{Z}'(q) \stackrel{\text{def}}{=} M\boldsymbol{Z}(q)$ as in Thm. C.1. We define the automaton $\mathcal{A}'' = \big(\Sigma, Q'', \delta'', \lambda'', \rho''\big)$ analogously to $\mathcal{A}'$:

    (i) $Q' = Q \times \{0, \ldots, N\}$,

    (ii) $\delta'': ((q, i), \mathsf{a}, (q', i)) \mapsto \frac{1}{\boldsymbol{Z}'(q)_{N-i}} \cdot \big(M(\mathcal{L}_{\phi\delta}(q \xrightarrow{a/w} q') \otimes \boldsymbol{\beta}_\phi(q'))\big)_{N-i}$,

    (iii) $\lambda'': (q, i) \mapsto \mathbb{1}\{q = q_\iota, i = N\}$,[7]

    (iv) $\rho'': (q, i) \mapsto \frac{1}{\boldsymbol{Z}'(q)_{N-i}} \mathcal{L}_\rho(q)_{N-i}$.

**Theorem C.3** (Constrained String Sampling (2)). *We have that*

$$\mathbb{P}(\boldsymbol{\Pi}(q_\iota) = \boldsymbol{\pi} \mid |\boldsymbol{\Pi}(q_\iota)|_\phi \geqslant N) = \mathbb{P}(\boldsymbol{\Pi}''(q_\iota) = \boldsymbol{\pi}) \tag{52}$$

---

[7]As in $\mathcal{A}'$, one can start in $(q_\iota, n)$ to sample from $\mathbb{P}(\boldsymbol{\Pi}(q_\iota) = \boldsymbol{\pi} \mid |\boldsymbol{\Pi}(q_\iota)|_\phi \geqslant n)$.

*Proof.* By definition, we have the following equalities

$$\delta''((q,i), \mathsf{a}, (q',i)) \overset{\text{def}}{=} \frac{1}{\boldsymbol{Z}'(q)_{N-i}} \cdot \left(\mathcal{L}_{\phi_\delta}(q \xrightarrow{a/w} q') \circledotimes \boldsymbol{\beta}_\phi(q')\right)_{N-i} \tag{53a}$$

$$= \frac{1}{\boldsymbol{Z}'(q)_{N-i}} \cdot w \cdot \left(\boldsymbol{\beta}_\phi(q')\right)_{N-i-\phi_\delta(q \xrightarrow{a/w} q')} \tag{53b}$$

$$\rho''(q,i) \overset{\text{def}}{=} \frac{1}{\boldsymbol{Z}'(q)_{N-i}} \cdot \left(\mathcal{L}_\rho(q)\right)_{N-i} \tag{53c}$$

$$= \frac{1}{\boldsymbol{Z}'(q)_{N-i}} \cdot \mathbb{1}\{N - i \leqslant \phi_\rho(q)\}\rho(q). \tag{53d}$$

This allows us to show that the next-state probabilities again sum to $1$ as in Thm. 4.5. Further, to compute the probabilty of the path $\boldsymbol{\pi} = q_\iota \xrightarrow{a_1/w_1} \cdots \xrightarrow{a_m/w_m} q_m$, we again use the same telescoping product as in Thm. 4.5. ∎

### C.1 RUNTIME OF CONSTRAINED SAMPLING

We analyze the runtime of constrained string sampling based on Thms. 4.4, 4.5 and C.3 as follows. Suppose we want to sample $K$ strings from a WFSA $\mathcal{A}$. We first determine the distribution of occurrences in each string using Thm. 4.4 or Thm. C.1 and then sample each string under said constraint with Thm. 4.5 or Thm. C.3.

(1) **Computing $\boldsymbol{Z}$**. $\boldsymbol{Z}$ can be computed with Lehmann's algorithm in $\mathcal{O}(|Q|^3 C)$ time for any WFSA over a compatible (complete) semiring, where $C$ depends on the complexity of the semiring operations—in our case, $\mathcal{O}(N + N^2)$ in a general base semiring and $\mathcal{O}(N + N \log N)$ using FFT over the reals.

(2) **Occurrence sampling**: We need to compute $\boldsymbol{Z}^{\otimes k}$ for $k \in \{0, \ldots, K\}$. This can be done in $\mathcal{O}(KN^2)$ or $\mathcal{O}(KN \log N)$ time depending on the semiring.

(3) **Event occurrence sampling**: This corresponds to constructing the WFSA $\mathcal{A}'$ and sampling from it. Since $\mathcal{A}'$ is a WFSA with $\mathcal{O}((|Q|N))$ states and $\mathcal{O}(|\Sigma|(|Q|N)^2)$ transitions, we can construct it in $\mathcal{O}(|\Sigma|(|Q|N)^2)$ time.

Taken together, the constrained sampling is dominated by the runtime of the first two steps, which is $\mathcal{O}(\max(K, |Q|^3)N^2)$ or $\mathcal{O}(\max(K, |Q|^3)N \log N)$ depending on the semiring. In practice, we find this approach to be two orders of magnitude faster than rejection sampling.

## D TARGETED KL DIVERGENCE VIA DECOMPOSITION

We are interested in measuring the impact of learnability on targeted features. To this end, we derive a decomposed KL divergence that works on the transition, symbol, or state level between an automaton and a trained LM.

Let $p_\mathcal{A}$ be an LM defined by a DPFSA $\mathcal{A}$ over states $Q$ and symbols $\Sigma$. Any string $x$ sampled from $p_\mathcal{A}$ decomposes into transitions consisting of states $q$, symbols $\sigma$, and weights $w$. Given another LM $p_\theta$ and a set of transitions of interest $\mathcal{T} \overset{\text{def}}{=} \{\delta \mid \phi_\delta(\delta) \neq 0\}$, for some event function $\phi$, we decompose the KL divergence to analyze how well $p_\theta$ captures these properties. At each step, $p_\mathcal{A}$ takes transition $\delta$ with probability $w$, while $p_\theta$ predicts the next symbol given the history $\boldsymbol{\sigma}_{<i}$ of all symbols preceding position $i$.

Throughout, let

$$\pi(q) \overset{\text{def}}{=} \Pr_{x \sim p_\mathcal{A}} [\delta(x) = q], \qquad \pi(q,a) \overset{\text{def}}{=} \pi(q)p_\mathcal{A}(a \mid q), \qquad \pi(a) \overset{\text{def}}{=} \sum_{q \in Q} \pi(q,a). \tag{54}$$

We have that

$$D_{KL}(p_{\mathcal{A}} \| p_\theta) = \mathbb{E}_{\boldsymbol{\sigma} \sim p_{\mathcal{A}}}\Big[\log \frac{p_{\mathcal{A}}(\boldsymbol{\sigma})}{p_\theta(\boldsymbol{\sigma})}\Big] \tag{55a}$$

$$= \sum_{\boldsymbol{\sigma} \in \Sigma^*} p_{\mathcal{A}}(\boldsymbol{\sigma}) D_{KL}\big(p_{\mathcal{A}}(\cdot \mid \boldsymbol{\sigma}) \| p_\theta(\cdot \mid \boldsymbol{\sigma})\big) \qquad \text{(chain rule on prefixes, 55b)}$$

**State-wise decomposition.** Group every history by the state it reaches, $q = \delta(\boldsymbol{\sigma})$:

$$D_{KL}(p_{\mathcal{A}} \| p_\theta) = \sum_{q \in Q} \sum_{\substack{\boldsymbol{\sigma} \in \Sigma^* \\ \delta(\boldsymbol{\sigma})=q}} \pi(\boldsymbol{\sigma}, q) D_{KL}\big(p_{\mathcal{A}}(\cdot \mid q) \| p_\theta(\cdot \mid \boldsymbol{\sigma})\big) \qquad \text{(insert } 1[\delta(\boldsymbol{\sigma}) = q], \text{56a)}$$

$$= \sum_{q \in Q} \pi(q) \underbrace{\sum_{\substack{\boldsymbol{\sigma} \in \Sigma^* \\ \delta(\boldsymbol{\sigma})=q}} \frac{\pi(\boldsymbol{\sigma}, q)}{\pi(q)} D_{KL}(p_{\mathcal{A}}(\cdot \mid q) \| p_\theta(\cdot \| \boldsymbol{\sigma}))}_{\text{Unweighted per-state contribution}} \tag{56b}$$

**Transition-wise decomposition.** Insert the next symbol $a$ and write $\pi(q, a) = \pi(q)p_{\mathcal{A}}(a \mid q)$:

$$D_{KL}(p_{\mathcal{A}} \| p_\theta) = \sum_{q \in Q} \pi(q) \sum_{a \in \Sigma} p_{\mathcal{A}}(a \mid q) \log \frac{p_{\mathcal{A}}(a \mid q)}{p_\theta(a \mid \boldsymbol{\sigma})} \tag{57a}$$

$$= \sum_{(q,a) \in \Delta} \pi(q, a) \underbrace{\log \frac{p_{\mathcal{A}}(a \mid q)}{p_\theta(a \mid \boldsymbol{\sigma})}}_{\text{Unweighted per-transition contribution}} \tag{57b}$$

$$\tag{57c}$$

**Symbol-wise decomposition.** Marginalise over states to obtain the symbol prior $\pi(a) = \sum_{q \in Q} \pi(q, a)$:

$$D_{KL}(p_{\mathcal{A}} \| p_\theta) = \sum_{a \in \Sigma} \pi(a) \mathbb{E}_{\boldsymbol{\sigma} \sim p_{\mathcal{A}}}\Big[\log \frac{p_{\mathcal{A}}(a \mid \delta(\boldsymbol{\sigma}))}{p_\theta(a \mid \boldsymbol{\sigma})}\Big] \tag{58a}$$

$$= \sum_{a \in \Sigma} \pi(a) \underbrace{\sum_{\boldsymbol{\sigma} \in \Sigma^*} \frac{p_{\mathcal{A}}(\boldsymbol{\sigma})p_{\mathcal{A}}(a \mid \delta(\boldsymbol{\sigma}))}{\pi(a)} \log \frac{p_{\mathcal{A}}(a \mid \delta(\boldsymbol{\sigma}))}{p_\theta(a \mid \boldsymbol{\sigma})}}_{\text{Unweighted per-symbol contribution}}. \tag{58b}$$

# E  EXPERIMENTAL SETUP DETAILS

**Sampling PFSAs.** We begin all our experiments by sampling PFSAs. The sampling procedure of a single automaton $\mathcal{A}$ is as follows: For each source state $q \in Q$, we sample a set of symbols, $\sigma \in \Sigma$, where each symbol has $0.5$ chance of being included. We then randomly sample a target state $q' \in Q$ for each symbol. This gives us a set of unweighted transitions between states and associated symbols. With probability $0.3$, we set each state to have a nonzero final weight, where we sample the state's outgoing and final weights from the Dirichlet distribution.

**Intervention Sampling.** We can conduct three types of causal interventions: on the number of times a certain **symbol**, or **state**, is seen during training. These interventions are best described with $\mathrm{do}$-notation introduced in §3. Each of these is captured by some property $P$, by intervening on it we sample according to $\mathcal{D} \sim \mathbb{P}(\cdot \mid \mathrm{do}(P = p))$. For symbol intervention on some symbol $a_I$, this corresponds to the even function $\phi_{a_I}(q, a, r) \stackrel{\text{def}}{=} \mathbb{1}\{a = a_I\}$. For transition interventions on transition $\delta_I$, $\phi_{\delta_I}(q, a, r) \stackrel{\text{def}}{=} \mathbb{1}\{(q, a, r) = \delta_I\}$. For state interventions on state $q_I$, we use the function $\phi_{q_I}(q, a, r) \stackrel{\text{def}}{=} \mathbb{1}\{q = q_I\}$.

**Ancestral Sampling.** For ancestral sampling, we start the process from the initial state $q_\iota$. We then sample the next symbol by recursively selecting the transitions according to the PFSA's probability distribution. For example, given a state $q$ and the transition $q \xrightarrow{a/w} r$, the conditional probability of sampling $a$, i.e. probability $\mathbb{P}(a \mid q)$, is $w$.

**Neural Language Models.** We conduct experiments using both LSTM and Transformer-based LMs. The configuration of the neural LMs, including specific hyperparameters is given in App. F.

**Error bars** All error bars are the closed-form standard error as reported over the relevant grouped population in each experiment. The error is reported over the bin for the binned experiments, i.e., not averaged.

## F MODEL AND TRAINING DETAILS

We now detail the hyperparameters used for training both the RNN and Transformer models. While they share common settings such as batch size and optimizer, they differ in architectural design and training specifics. The loss we use is over the full KL divergence at each symbol position in the training data against the corresponding automaton, this can be seen as a sample-efficient distillation of the sampled automata.

We use a parameter budget of 128k for both LSTM and Transformers and train all models on CPU. For layer norm, we initialize weights to 1 and biases to 0. When dropout is applicable, we use a value of 0.1. Parameters are uniformly initialized from $[-0.1, 0.1]$. We shuffle data at training, with a max batch size of 128 tokens, and a learning rate of 0.01. We use the Adam optimizer, and clip gradients with a threshold of 5 using $L^2$ norm scaling. The learning rate is multiplied by 0.5 after 5 checkpoints with no decrease in the loss. Training is halted after 10 checkpoints if no improvement is observed.

Sampling of an individual dataset typically takes within minutes on a V100 32 GB GPU, as we make use of a GPU for the allsum calculations. The training of an individual dataset typically runs in minutes on a single modern CPU core using the LSTM architecture, while the Transformer models can run longer. Our large-scale experiments were run on a cluster with 76 nodes, making use of 16 CPU cores on each node, and occasionally on a single machine with 2 x AMD EPYC 9354 32-Core Processor and two NVIDIA H100. The parity+free experiment and the varying topology experiment run in under 12 hours, making use of the full CPU/GPU resources for both sampling and training. While, the larger-scale run took a few days to sample and train in parallel.

## G ADDITIONAL NOTES ON THE SCOPE OF THIS WORK

The goal of this paper is not to evaluate architectures nor do an exhaustive comparison of architectures, such as Mamba Gu & Dao (2024) or CoT Wei et al. (2022). The choice of LSTMs and Transformers is arbitrary. These are simply existence witnesses, toy use cases to demonstrate the main theoretical contribution. The key contributions are the causal framework and the sampling procedures. As such, this is not an empirical work, but a theoretical work. Furthermore, the choice of deriving and focusing only on the decomposed KL App. D as a measure of learnability is principled and exactly meets the needs of supporting the framework and justifying it through the use cases. Finally, the choice of focusing on DPFSAs is typical of this line of work; it encompasses all of the regular languages—languages that can encode a wide range of phenomena that are of value to the theoretical and empirical study of what neural language models can learn.

## H ADDITIONAL FIGURES

An additional figure for the at-least-once sampling variant of the machine in Fig. 1 is given in Fig. 6.

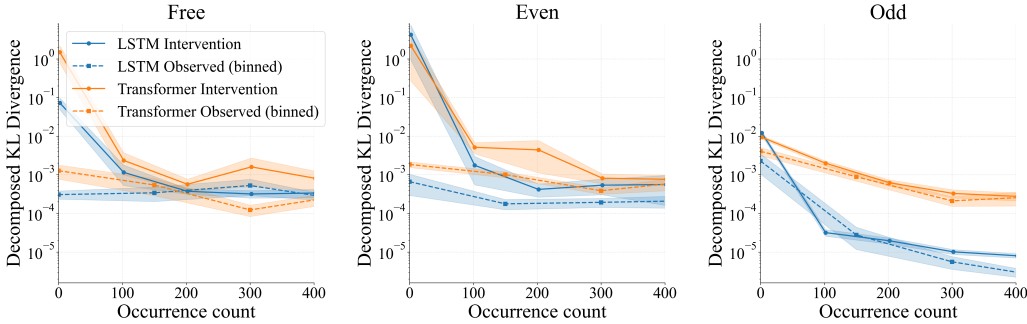

Figure 6: Monte Carlo estimates of the state-wise learnability for the automaton given in Fig. 1 using the at-least-once intervention.

