# OpenReview forum: "Measuring Scarcity–Complexity Collision in Language Model Estimation"
_ICLR.cc/2026/Conference — Submitted to ICLR 2026_

### Official Review · Reviewer_poG2 · 2025-10-20

**Soundness:** 3
**Presentation:** 2
**Contribution:** 2
**Rating:** 4
**Confidence:** 3

**Summary:**

This paper addresses the limitations of learning in neural language
models from two perspectives: scarcity and complexity.  While the
limitations caused by scarcity can be mitigated by increasing
sampling, those arising from complexity cannot.  Using Pearl’s causal
framework of do-calculus, the authors propose a scarcity measure,
defined as a causal quantity over event-occurrence distributions on a
probabilistic finite-state automaton (PFSA).  In other words, rather
than simply counting how rare an element (such as a state, transition,
or symbol) is, the measure captures the effect of scarcity by
observing how the model’s performance changes when the occurrence
frequency of that element is intervened upon in the sense of a
do-operation.


In the experimental section, the authors apply this framework to
controlled language-learning tasks implemented on probabilistic
finite-state automata.  They train neural sequence models on data
sampled both observationally (following the natural frequency
distribution) and interventionally (where event frequencies are
artificially balanced).  By comparing the resulting state-wise
decomposed KL divergence, the study demonstrates that some states
remain difficult to learn even when their frequency is increased,
revealing structural or complexity-driven limitations.  This provides
empirical evidence that the proposed scarcity measure successfully
distinguishes between performance drops caused by data scarcity and
those stemming from intrinsic structural complexity.

**Strengths:**

1.  The paper makes a rigorous and original distinction between two often conflated sources of learning difficulty between
(a) scarcity — limitations due to insufficient sampling of certain events, and
(b) complexity — limitations intrinsic to the structural or grammatical properties of the system.
This conceptual clarity is a contribution to the theory of learning and generalization.

2.  By grounding the definition of scarcity in Pearl’s causal framework (via do-calculus), the authors move beyond conventional frequency-based or statistical interpretations. Defining scarcity as a causal quantity — i.e., the performance change under an intervention on event frequencies — is both elegant and generalizable. This gives the notion of “data scarcity” a principled mathematical meaning.


3.  The use of probabilistic finite-state automata (PFSA) as a controlled experimental environment is a major methodological strength.
It allows the authors to isolate structural dependencies and control for event frequencies in a way that is impossible with natural-language corpora. The PFSA setup serves as a minimal model of linguistic learning, bridging formal language theory and neural estimation.


4.  The comparison between observational and interventional training distributions provides intuitive yet rigorous evidence.
The results (e.g., decomposed KL divergence across states) demonstrate when poor learning performance is due to low frequency versus when it arises from structural complexity.
This empirical clarity strengthens the theoretical claims.


5.  The framework establishes a foundation for future studies on causal interpretability in machine learning.
By quantifying how much of a model’s limitation is attributable to data availability versus intrinsic structure, it contributes to ongoing debates about scaling, generalization, and data efficiency in neural models.

**Weaknesses:**

1. The empirical validation is conducted only on synthetic data generated
by probabilistic finite-state automata (PFSA).
While this controlled setup isolates causal effects cleanly, it leaves
open whether the proposed scarcity measure generalizes to
real-world learning tasks.
No results are provided for larger neural architectures or open-domain data.


2. The causal definition of scarcity, though elegant, remains
mathematically abstract and difficult to operationalize beyond the
PFSA framework.
The paper does not fully specify how the intervention on event
frequencies could be implemented for arbitrary datasets or continuous
domains.


3. Complexity is treated largely as structural difficulty in the
automaton (topological or dependency-based).
However, other important aspects of complexity in real world data
are not captured in this formulation. It goes largely beyond PFSA.


4. The causal estimation procedure requires retraining models under multiple interventions (different sampling distributions), which can be computationally expensive.
The feasibility of this approach for large-scale neural systems remains untested.


5. The paper positions itself mostly within formal and causal theory, but
provides limited comparison to prior empirical findings on data
imbalance, curriculum learning, or frequency effects.
As a result, its impact on practical neural-language modeling remains
somewhat speculative.

**Questions:**

1.  This framework is limited to probabilistic finite-state automata
(PFSA).  The significance of neural networks lies in their ability to
learn patterns that go beyond PFSA, encompassing higher levels of the
Chomsky hierarchy.  How robust is the causal definition of scarcity
when extended beyond PFSA formalisms?


2. Can the scarcity measure be generalized to non-discrete settings?


3. How interpretable is the scarcity measure numerically?

4. Can this framework inform training strategies?

---

> ### Author Response · Authors · 2025-11-14
>
> Thanks for the review and the highlighting of the many strengths. We address your comments one by one below.
>
> ### 1. The empirical validation is conducted only on synthetic data generated by probabilistic finite-state automata (PFSA).
>
> The study of language models learnability with languages from formal automata is an established field as highlighted in the introduction and in the related work in our paper. Our methodology is exactly for these settings.
>
> ### 2. The causal definition of scarcity, though elegant, remains mathematically abstract and difficult to operationalize beyond the PFSA framework. The paper does not fully specify how the intervention on event frequencies could be implemented for arbitrary datasets or continuous domains.
>
> First, we see no inherent need to operationalize the tooling beyond the PFSA setting, it does what it needs to do in the setting we propose–which is studied widely. We do claim however, that the semiring can be applied for other probabilistic settings, such as if the automata were allowed to read or write from a tape or stack. Though we leave this to future work. Does this answer your comment? Is this what you meant with arbitrary datasets – that they would be higher up in the chomsky hierarchy? In general going higher up is not trivial.
>
> ### 3. Complexity is treated largely as structural difficulty in the automaton (topological or dependency-based). However, other important aspects of complexity in real world data are not captured in this formulation. It goes largely beyond PFSA.
>
> We separate whether a model performs worse due to the probability of seeing the event, as opposed to the event as such being hard to model. It is this second aspect that we refer to as complexity. The scope of our study is the learnability of languages defined by PFSAs, a class commonly used to analyze language models learnability. We do not claim that complexity does not exist outside PFSAs. We are only considering formal languages precisely because that gives the laboratory setting that allows researchers to rely on the tools of classical computer science to understand language models.
>
> ### 4. The causal estimation procedure requires retraining models under multiple interventions (different sampling distributions), which can be computationally expensive. The feasibility of this approach for large-scale neural systems remains untested.
>
> This is how learnability of language models using formal automata is studied. The architectures used are small and we make the sampling efficient. If this was done at a grand scale (billion parameter models) it would be expensive, but that does not mean that it is not useful to do it at all—as evidenced by the wide range of work we cite. Having said that, what was once large is now small, so who knows.
>
> ### 5. The paper positions itself mostly within formal and causal theory, but provides limited comparison to prior empirical findings on data imbalance, curriculum learning, or frequency effects. As a result, its impact on practical neural-language modeling remains somewhat speculative.
>
> We are providing methodology for evaluation of language models learnability of formal languages. We are not claiming to make empirical contributions with regards to specific claims about architectures, we are making a point about how one should evaluate those. Comparisons to some particular curricula are thus not relevant. The impact on practical neural-language modeling will be made from having the proper causal tools to evaluate the architectures. People are already doing this, we are showing them better tools.
>
> ### 5. This framework is limited to probabilistic finite-state automata (PFSA). The significance of neural networks lies in their ability to learn patterns that go beyond PFSA, encompassing higher levels of the Chomsky hierarchy. How robust is the causal definition of scarcity when extended beyond PFSA formalisms?
>
> The causal definition is exactly the same. We give the causal model independently of the formal machinery that can be used to realize it in the PFSA setting. As long as one can intervene on a property directly the causal model applies.
>
> ### 6. Can the scarcity measure be generalized to non-discrete settings?
>
> Any measurement could be applied as long as it is relevant. Without going into details on this hypotheticas one can imagine using MSE over a masked area in an image or something of that sort. This point is a bit unclear to us.
>
> ### 7. How interpretable is the scarcity measure numerically?
>
> The decomposed KL allows us to measure exactly how the distribution being modeled differs from the ground truth when the model is faced with the specific property. This makes it very targeted. The KL interpretation is the standard one.
>
> ### 8. Can this framework inform training strategies?
>
> By better estimating learnability of the training strategy then yes, it can shed light on how well a given strategy works.
>
>
> We are happy to elaborate on any point.

---

> > ### Comment · Reviewer_poG2 · 2025-11-21
> > **Thank you for your comment**
> >
> > Thank you very much for your reply. Your reply clarified my concerns. I'll update my scores once seeing all the discussions.

---

> > > ### Author Response · Authors · 2025-11-26
> > >
> > > Thank you for acknowledging our reply. We are happy the reply clarified your concerns!

---

### Official Review · Reviewer_Cemk · 2025-10-30

**Soundness:** 2
**Presentation:** 1
**Contribution:** 2
**Rating:** 4
**Confidence:** 3

**Summary:**

This paper introduces a causal framework to disentangle scarcity (low frequency of a linguistic pattern in training data) from complexity (inherent difficulty of learning the pattern for a given architecture) as causes for poor language model performance. The authors use Probabilistic Finite-State Automata (PFSAs) as a controlled testbed. Their main contributions are: 1) A formal causal model for LM learnability based on Pearl's do-calculus, and 2) An efficient "binning semiring" algorithm for sampling data from PFSAs under interventions that fix the frequency of specific events (symbols, states, transitions). They illustrate the framework with case studies on LSTMs and Transformers, showing that correlational analyses can be misleading and that interventional estimates can invert conclusions about what is hard to learn.

**Strengths:**

- The authors go to great length to carefully disentangle what they identify as two confounders in the learning difficulties of transformers, scarcity and complexity.
- The ability to causally attribute LM failures to data or architecture is significant. It provides a more principled methodology for failure analysis and dataset design in synthetic settings, moving beyond correlational claims.

**Weaknesses:**

- **Clarity**: Speaking as a theorist myself, I found that the formalism, particularly in Sections 4 and the appendix, is heavy. The connection between the complex binning semiring machinery and the high-level goal of "making rare events less rare" feels disproportionate. This complexity obscures the intuition and makes the paper difficult to digest. In some places, the writing appeared convoluted; for example, the caption for Fig. 4 refers to "Monte Carlo estimates" - did the authors simply sample several data sets and train a couple of models on them, or was there something more going on?

- I also had some difficulty with some **conceptual issues**. The task that the authors actually focus on is essentially the parity task, which is a nice way of testing the ability of a network to do something akin to counting, a well-known issue for transformers. Now if you try to learn the parity function (or any function) and you are in the right function class (i.e. your architecture is expressive enough to even represent the task), the fundamental difficulty of a task that remains is its sample complexity -- how many samples will I need to learn the function? Parity is a popular test bed for computational complexity because it is hard to learn, as we know since Minsky & Papert in the 1960s; more recently, arXiv:2207.08799 by Boaz Barak et al. gave a nice discussion. In the present paper, the authors seem to distinguish scarcity (i.e. a training set below the required sample complexity) from "complexity", but I am not sure what complexity means here? The authors seem to intend it to mean some inherent difficulty for certain architectures - so that would be simply a higher sample complexity? (In fact, since transformers do not have a hidden state, contrary to LSTMs, I would expect them to have a higher sample complexity to learn the parity)

**Questions:**

- Please clarify the distinction between complexity and scarcity (see my point above)

- Relation to other hard to learn functions: How does your work compare to the setting where we try to learn a sparse parity function? What does the formulation via formal languages add? Could you apply your causal machinery to that case?

- Intervention Specifics: In the case studies, when you perform an intervention like "symbol a occurs exactly N times," how is this intervention not also changing the language that you are learning? You introduce a mismatch between training and test distribution here. I understand that this is for the benefit of your causal intervention, but I think it's worth discussing how that impacts the language from which you are now effectively sampling the training set.

---

> ### Author Response · Authors · 2025-11-14
>
> Thank you for the review, we answer your comments and questions below.
>
> ### 1. I found that the formalism, particularly in Sections 4 and the appendix, is heavy. The connection between the complex binning semiring machinery and the high-level goal of "making rare events less rare" feels disproportionate.
>
> We acknowledge the formalism can be seen by some as somewhat heavy, but at the same time we think there is a need for some detail to get the sampling methods right—we will clarify our position on this. For instance, some mathematicians might find it enough to say ‘quotient semiring over polynomials by the ideal over polynomials of degree $N+1$’ but we wanted to expose the internals of these operations to build up a more transparent view of the sampling procedure. From reading the other reviews we will clarify this motivation and ensure that bigger picture and the need for the formalities are better justified. Does this answer your comment or do you feel there is something in particular that feels disproportionate?
>
> ### 2. The caption for Fig. 4 refers to "Monte Carlo estimates" - did the authors simply sample several data sets and train a couple of models on them, or was there something more going on?
>
> We agree that “Monte Carlo estimates” is not strictly needed in the figure but the “Monte Carlo” phrase has a specific meaning in that we are estimating the equations in figure 3. That is, we sample automata from some family of machines that we wish to study the property under, since we do not have the means to tractably derive the probabilities we use sampling (Monte Carlo) to get an estimate.
>
> ### 3. The task that the authors actually focus on is essentially the parity task, which is a nice way of testing the ability of a network to do something akin to counting, a well-known issue for transformers.
>
> We would like to make a clarifying comment here. We only use the parity task as one of our example settings since it is well studied and, and it allows us to demonstrate the learnability measurement (the targeted decomposed KL). The results we show are known beforehand, and since it's a simple language we find it particularly interesting to see that causal intervention does show different trends than observational at the accepting even state. This last point stresses, that if one wishes to know the learnability of an architecture for a property, then one should do so causally, otherwise the results will differ.
>
> ### 4. In the present paper, the authors seem to distinguish scarcity (i.e. a training set below the required sample complexity) from "complexity", but I am not sure what complexity means here? The authors seem to intend it to mean some inherent difficulty for certain architectures - so that would be simply a higher sample complexity?
>
> By intervening on the generative process we can force a property to happen more often in the sampled data than it would do otherwise. For instance, consider a classic example from natural language: “Bananas are yellow”. Since everyone knows bananas are yellow nobody says it and thus it is rare to see the trigram written in text. There is however nothing complicated about this sentence. If a model has failed to learn that bananas are yellow it is simply because there is not enough data. Here, scarcity of the property can explain that it is not learned. By intervening on the process behind the data we could ask someone to explain bananas in more detail and the model would have enough signal. Not every example is like this however, as with the parity problem, even if we force a lot of such examples to appear in our data the Transformer will still struggle. This is the difference, the proposed method separates these cases. The complexity is thus sample complexity, assuming it can be learned eventually, or just that it is too complex for the setting. We do not claim to separate the types of complexity, only to say whether it is due to the low probability of seeing the event in the data (scarcity) or since it is hard or impossible to learn (complexity). We will clarify this in the updated paper. Does this answer your question?
>
> ### 5. Since transformers do not have a hidden state, contrary to LSTMs, I would expect them to have a higher sample complexity to learn the parity.
>
> We agree, recurrent models can count since they maintain a state unlike the standard Transformer. The main point in our paper is not study this property however, this is mainly an example to demonstrate our method, and we use these examples (and their known properties) to validate and justify it—by showing that the numbers we see change.

---

> > ### Author Response · Authors · 2025-11-14
> >
> > ### 6. Relation to other hard to learn functions: How does your work compare to the setting where we try to learn a sparse parity function? What does the formulation via formal languages add? Could you apply your causal machinery to that case?
> >
> > Sparse parity (deciding whether there is an even number of 1s in some fixed positions rather than in the full sentence) can be encoded as a DFA and the tooling could thus be applied. To see this we can encode the machine to keep track of the current position and only consider flipping the parity in these positions. The machine will however grow in number of states with the number of positions needed to keep track of (the maximum index of the state we wish to consider the parity for). Does this answer your question?
> >
> > ### 7. When you perform an intervention like "symbol a occurs exactly N times," how is this intervention not also changing the language that you are learning?
> >
> > We think we see where you are coming from here. But the change is surgical in the sense that it only changes the joint distribution of the languages features for the specifically targeted property. In that sense it allows us to ask the counterfactual, “how would the distribution over the language look if the probability of this event was different”? By training on such data and evaluating the targeted learnability (the decomposed KL) we can ask for the property of interest whether it was learned differently when we force it to occur a specific number of times or not. Since we are interested in understanding the targeted property itself (and we average over many languages to analyze it, either using different weights, transitions, states etc.) the change in the language is not a worry.
> >
> > If there is something still unclear we’d be happy to elaborate on any of the points above.

---

> > > ### Comment · Reviewer_Cemk · 2025-11-26
> > >
> > > Thank you for your detailed reply to my comments.
> > >
> > > > We only use the parity task as one of our example settings since it is well studied [...] The results we show are known beforehand [...] we find it particularly interesting to see that causal intervention does show different trends than observational at the accepting even state. This last point stresses, that if one wishes to know the learnability of an architecture for a property, then one should do so causally, otherwise the results will differ.
> > >
> > > It's perfectly fine to use the parity task as an example, indeed it's a good idea since there is a range of results to compare. My problem with this paper, that I also see reflected in some other reviews, is I still do not understand what "different trend" this paper is observing. Assuming that both the transformers and LSTMs that you use can learn the parity (otherwise this exercise seems pointless to me), do you show different sample complexities for the models with respect to each other? under different input distributions? how do your results compare to results from Barak et al. ? etc
> > >
> > > > By intervening on the generative process we can force a property to happen more often in the sampled data than it would do otherwise. For instance, consider a classic example from natural language: “Bananas are yellow”. Since everyone knows bananas are yellow nobody says it and thus it is rare to see the trigram written in text. There is however nothing complicated about this sentence. If a model has failed to learn that bananas are yellow it is simply because there is not enough data. Here, scarcity of the property can explain that it is not learned. By intervening on the process behind the data we could ask someone to explain bananas in more detail and the model would have enough signal. Not every example is like this however, as with the parity problem, even if we force a lot of such examples to appear in our data the Transformer will still struggle. This is the difference, the proposed method separates these cases. The complexity is thus sample complexity, assuming it can be learned eventually, or just that it is too complex for the setting. We do not claim to separate the types of complexity, only to say whether it is due to the low probability of seeing the event in the data (scarcity) or since it is hard or impossible to learn (complexity). We will clarify this in the updated paper. Does this answer your question?
> > >
> > > Thank you for trying to find another way to explain your point. I do have a couple of objections though. First of all, I think the example is not a great choice - there is plenty of text around that says bananas are yellow, and language models are good at this type of recall. But okay, let's assume for a moment that this information was too scarce in the data set. Sure, increasing its frequency will help the model learn this mapping. But this is very different from the parity example: in parity, the difficulty of the task is a consequence of the strong influence each element of the input has on the output - changing any element of the input changes the output.
> > >
> > > In summary, my main concern - that there are conceptual issues with the notions of complexity and scarcity as used in this paper - remains, and so I maintain my score.

---

> ### Author Response · Authors · 2025-11-26
>
> > I still do not understand what "different trend" this paper is observing.
>
> The *trend* we say is different is the **difference in the plots when the data is sampled under intervention or obtained correlationally**. If you look at Figure 2 for the parity+free machine, notice how the learnability is significantly higher (lower decomposed KL) for the observational results rather than the interventional results at the Even state. Note that the y-axis is log scale. For both the Transformer (orange) and the LSTM (blue), the results differ between the causal and correlational settings. This is the key point in general: if we don't sample causally, we will see a different relationship/trend between the learnability (decomposed KL divergence) and the occurrence count.
>
> Consider the even state: the difference in sample complexity between the LSTM and Transformer is directly readable from the y-axis. Given the same amount of data, the learnability differs—this is the difference in complexity. We can make it formal by saying for some $\epsilon>0$, how many steps do we need to get the KL below $\epsilon$. Let’s say we set $\epsilon=0.001$, reading of Figure 2 for the Even state, we see:
>
> 1. LSTM + observation: hits $\epsilon$ after around 100 occurrences.
> 2. LSTM + intervention: We see it actually happens around 1000 occurrences.
> 3. Transformer + observation: around 500 occurrences needed.
> 4. Transformer + intervention: It barely gets there around 2000 occurrences.
>
> > Assuming that both the transformers and LSTMs that you use can learn the parity (otherwise, this exercise seems pointless to me), do you show different sample complexities for the models with respect to each other? under different input distributions?
>
> The transformer can not learn parity without modifications as mentioned in the paper, see e.g., Hahn & Rofin 2024. But this does not matter. The point is that we see different results under intervention and observation, and the learnability as a function of the number of occurrences differs (the sample complexity trend) for the two architectures, as is known.
>
> > How do your results compare to results from Barak et al. ?
>
> The discussion in Barak et al is nice and we will add it as a general citation on parity, but the point we are making is not about the parity language *as such*—we are simply using it as an example and we could have made the comparison using other languages, **the key point is that the observational trend is different from the causal trend**, we are not making any claims about parity or the difference in Transformers and LSTMs, merely observing that we do see the same trends as are already known.
>
> > The banana example is very different from the parity example: in parity, the difficulty of the task is a consequence of the strong influence each element of the input has on the output - changing any element of the input changes the output.
>
> The connection to the banana example is the value of $\eta$ in Figure 1 (the machine that encodes the two tasks: the starfree language and the parity language); we are not comparing it to the parity language on its own.
>
> If $\eta$ is low, we will have few strings of the form $aa*$ in the sampled dataset, and the parity examples will be more common. If we knew nothing about the learnability of the $aa*$ task then there would be two potential reasons for the model performing well on the task; it not being learned since $\eta$ is low (this is the scarcity) or because the task is too complex (even if we increase probability of seeing the task specific data (star free samples) we don’t get the targeted learnability below some $\epsilon>0$). Similarly, if $\eta$ is high, we will have few examples of the other task encoded by the machine, the parity language. This is the whole difference between scarcity and complexity. By enabling causal intervention, we can plot the true relationship between learnability and the number of occurrences to find the intersection of the learnability with a line drawn through $\epsilon$ on the y-axis.
>
> To further illustrate this, consider the sketch in https://i.imgur.com/MtEnZ1j.png. Here, the purple circle corresponds to a specific observation. For this hypothetical task, if we only look at the observation, we don’t know whether the task is not learned because the task is complex (we require many examples to get below $\epsilon$ or if decreasing the scarcity (the probability of seeing the task in the training data) will get us below $\epsilon$. The interventional approach lets us sample exactly the points we need to efficiently get the blue line and settle the matter.
>
> We could collect many observational points as we do in our experiment, but without the intervention control, we have to collect very many such points to observe a trend over a larger interval (which is often prohibitively expensive or impossible for low probability events). And, even after doing so, we show that the observational trend differs from the causal one.

---

### Official Review · Reviewer_xjHV · 2025-11-01

**Soundness:** 3
**Presentation:** 3
**Contribution:** 3
**Rating:** 8
**Confidence:** 3

**Summary:**

This work contributes a causal framework that evaluates the learnability of probabilistic formal languages. The core operation is a controlled sampling procedure that intervenes on the target event frequency. By controlling frequency, one can decouple two factors that contribute to the difficulty of learning a property: scarcity and complexity, which are hard to disentangle in an observational setting. The authors demonstrate the effects of the proposed framework through three case studies that analyze the learnability differences between the Transforms and LSTM architectures.

**Strengths:**

* This work proposes a rigorous causal framework that decouples scarcity and complexity through frequency interventions, allowing more fine-grained analysis on learnability. This could be a powerful analysis tool to better understand data and architecture biases.
* This work is well written. In particular, the problem and the motivation are clearly stated.
* Originality: As I do not work on formal languages, I am not familiar with the literature in this area, but this could potentially be a novel application of causal tools in analyzing LMs with formal languages.

**Weaknesses:**

* Based on Figure 2 and Figure 4, IIUC, while for a given architecture, the intervention results generally differ from observational results, the relative differences between the architectures seem relatively consistent across methods, i.e., for someone only interested in comparing the learnability of an architecture, comparing based on observational results would mostly yield same results as comparing based on intervention results. What are some cases where the learnability analysis would yield different results from the two methods?

* The work focuses on discussing one type of property P, i.e., strings that traverse a set of transitions exactly N times, what are some alternative properties that might be useful, and how would they change the sampling process?

**Questions:**

See weaknesses.

---

> ### Author Response · Authors · 2025-11-14
>
> Thanks for the positive review, we are glad you found the work clear and well written.
>
> ### 1. Based on Figure 2 and Figure 4, IIUC, while for a given architecture, the intervention results generally differ from observational results, the relative differences between the architectures seem relatively consistent across methods, i.e., for someone only interested in comparing the learnability of an architecture, comparing based on observational results would mostly yield same results as comparing based on intervention results. What are some cases where the learnability analysis would yield different results from the two methods?
>
> This is an interesting question in its own right, and not one that we specifically aim to answer as such. It is hard to say where exactly we would see a difference in two configurations/architectures up front but we can imagine a setting where model A is good at one language $L_1$, and model B is good at another language, $L_2$, and when they are combined such that one is overrepresented in the correlational setting, but not in the causal setting, then the results would flip.
>
>
> ### 2. The work focuses on discussing one type of property P, i.e., strings that traverse a set of transitions exactly N times, what are some alternative properties that might be useful, and how would they change the sampling process?
>
> Since we focus on finite state automata every property can be defined in terms of a set of transitions (these could be transitions with some symbol, transitions that are x-distance from an accepting state, transitions with loops, and so on). If we were to extend our approach to other events, such as reading from a tape or stack, then those events could also be counted. This would extend our work to more expressive languages, but we leave this setting to future work. Does this answer your question or did you have some different kind of property in mind?

---

### Official Review · Reviewer_vAd1 · 2025-11-02

**Soundness:** 2
**Presentation:** 1
**Contribution:** 2
**Rating:** 2
**Confidence:** 2

**Summary:**

This work identifies a key issues with evaluating capabilities in LMs: if a model is unable to perform some task, this may either be caused by the task being (a) low frequency (scarce) in the pre-training data, or (b) computationally complex, e.g. an NP-complete problem vs. a problem that can be solved by a simple DFA. The authors propose using formal languages as a domain to study this, since data frequency and computational complexity can be formally measured and systematically varied. They develop a method for generating formal language data which holds one of these factors - scarcity or complexity - fixed, while varying the other. They contextualize this work in a causal inference framework, contrasted with prior correlational work which confounds scarcity with complexity. With three empirical case studies using toy examples, the authors show cases where their causal framework makes opposite predictions compared with a correlational analysis.

I find the authors motivation and setup to be compelling. However, I found the writing and presentation to be hard to comprehend, and I had trouble making sense of how the authors results support their key points. The writing seems to alternate sharply between simple and straightforward (although a bit redundant), especially with the main points about scarcity vs. complexity, or dense and hard to understand with the methods and results. I think the writing and presentation could be improved significantly to smooth these transitions, and more clearly connect the authors technical contributions with their main arguments. If the presentation was improved to that I could better understand - at least to some extent (see following point) - specifics of the methods, results, and how the results support the authors' claims, I would raise my score at least to a 4.

I had trouble understanding section 4 due to my lacking relevant technical background, and so I am unable to provide feedback on the correctness of section 4 or the significance of this contribution.

**Strengths:**

- The scarcity-complexity problem is interesting and I haven't seen a paper which explicitly tries to study this point. I also appreciate the contrast with prior works with correlational analysis.
- The domain of formal languages seems like a good fit for systematically studying this problem.
- The authors' proposed method of generating data which holds either scarcity or complexity fixed while varying the other property, seems like an apt tool for studying this problem.

**Weaknesses:**

- I found the writing and presentation of their framework in sections 2 and 3 to be difficult to understand.
	- I could comprehend the key broader points, but I found the writing to be lacking in terms of connecting key ideas with the authors' math.
	- In Fig. 2, what should I be taking away from this figure? Why does it matter that there is alignment for Odd but not Free or Even? I had to flip back and forth between this figure and the paragraph in section 2 on the following page to make any sense of this, and I struggled to understand exactly how this result matches the authors' main point about scarcity vs. complexity.
	- In Fig. 3, I'm not sure what to make of the notation on the right hand side, e.g. what does P(dl | dd) mean? Is dl short for do(l)? The figure caption is very vague and didn't help me either with interpreting the figure, or with what I should take away when looking at it. What does it mean to "restores the marginal P(a)", and why should I care about this? Should I be concluding something about the advantage of intervening > conditioning? Do both of the equations on the right match the diagram on the left?
- I similarly found interpreting the results in section 5 to be difficult to understand. I think it would help if the key results were explained more clearly in terms of how they support the authors' primary claims.
	- E.g. "In the former, we see an interesting phenomenon: not only is there a clear difference in the decomposed KL, but for lower occurrence counts, there is an inverse trend indicating complex structural confounders for machines that naturally produce a low number of occurrences." I think the point here may be that correlational analysis can be misleading, and can show an inverse effect compared with interventional analysis, which is known to be correct.
	- Figs 4, 5 - again, I'm not sure how to interpret these results at a high level based on the captions, or what the key takeaways from these figures are. A particularly salient example is the sentence in caption for fig 5: "We see how observational results can lead to misleading assumptions." - *how* do observational results lead to misleading assumptions? What are those results and assumptions?

**Questions:**

- See questions in first bullet point for weaknesses
- I had trouble understanding the novel method proposed in section 4, although I lack formal math background and it could be that I'm just not the right reviewer to understand this paper. I am unable to provide feedback on the correctness of section 4 or the significance of this contribution.
- Relatedly, a general question I have is, who is the target audience for this work - what sort of technical background and fields of interest would you expect a reader to have? On one hand, I'm excited about the general problem framing, and the general setup seems interesting to me and I think I could understand the experiments and results if they were explained more clearly, but on the other hand, I was entirely lost in section 4.

---

> ### Author Response · Authors · 2025-11-14
>
> Thanks for the review!
>
> ### 1. Sections 2 and 3 to are difficult to understand and issues in connecting key ideas with the authors' math.
>
> In section 2 we introduce language models (following e.g. Formal Aspects of Language Modeling, Cotterell et al, 2023), and in section 3 we describe the causal graphical model (see e.g. Causal inference from graphical models by Lauritzen, 1999) and the two approaches (under intervention and not intervention) for the estimating the learnability of a property (when taken over many languages that exhibit it). We give some answers below on your more specific comments.
>
> ### 2. What should be taken away from figure 2? How does this result match the authors' main point about scarcity vs. complexity.
>
> The main point to be taken away from the figure is that its purpose is to validate our method: (a) We show that it can be used to recover known (and commonly referenced) results about the differences in what RNNs (such as LSTMs) and Transformers learn. (b) We see that the causal approach gives different results than using correlation. If one wants to establish the true learnability of the property the causal approach is needed. Does this answer your question?
>
> ### 3. In Fig. 3 what does P(dl | dd) mean? Is dl short for do(l)? What does it mean to "restores the marginal P(a)", and why should I care about this? Should I be concluding something about the advantage of intervening > conditioning? Do both of the equations on the right match the diagram on the left?
>
> The key takeaway is that when one follows the top equation (the correlational) one – the automaton (through the variable $da$) is dependent on the dataset having the property. This means that some languages are more likely to express the property so if we use this measurement we don't get an independent answer for how complex the property is. The bottom one, on the other hand, is intervened on such that the dependency goes away.
>
> The notation uses the $d$ symbol to indicate the pointwise (“measurable”) values in the sets (borel algebras) over Automata, Languages, and Datasets we consider, we will clarify the notation (you may recall it from calculus, measure theory or probability classes). So $da$ corresponds to an automaton,$dl$ to a language, and $dd$ a dataset.  “Restoring the marginal $P(a)$” means that we are averaging over all automata without conditioning them on the property we are investigating, which means that we are not biased towards seeing automata that are more likely to have the property to begin with.
>
> The equations are both derived from the diagram on the left, but when the intervention is made the edge from $A$ to $D$ is severed, this is the causal intervention.
>
> ### 4. "In the former, we see an interesting phenomenon..." I think the point here may be that correlational analysis can be misleading, and can show an inverse effect compared with interventional analysis, which is known to be correct.
>
> Yes, that is exactly the point – we will make this clearer. The main point is to say that correlational studies of learnability with formal languages (a common approach) will give biased results. And we introduce the tools for doing so causally.
>
> ### 5. Fig 5: "We see how observational results can lead to misleading assumptions." - how do observational results lead to misleading assumptions? What are those results and assumptions?
>
> Thanks, assumptions should have been 'conclusions'. The point is the same one as above—that a correlational study gives different results from a causal one.
>
> ### 6. I had trouble understanding the novel method proposed in section 4, although I lack formal math background and it could be that I'm just not the right reviewer to understand this paper. I am unable to provide feedback on the correctness of section 4 or the significance of this contribution.
>
> We want the paper to be understandable without grasping the full mathematics of the underlying machinery: Was it clear to you that the binning semiring combined with the sampling methods is what enables the interventions? That is, by intervening we can control the frequency (the `scarcity’) of a property), thus giving a causal result when evaluating learnability. We want to make sure this gets across.
>
> ### 7. What fields of interest would you expect a reader to have?
>
> The target audience is those who are interested in the evaluation of architectures and configurations and see value in using formal languages. Since we introduce elaborate tooling (the binning semiring and the causal approach) we feel some mathematical background (familiarity with automata over semirings, probabilitiy theory, causal models) is needed to fully understand it. We will clarify the motivation and are very appreciative of any feedback in this regard.
>
> Have these points helped to clarify the contribution and how we support it? We are happy to answer any further questions. Getting these kind of questions is great.

---

### Comment · Area_Chair_YEp8 · 2025-11-25
**Please discuss**

This paper has a wide range of scores. I would love to see the reviewers engage with each other, as well as the author response, and see if they come closer to a consensus.  Have the rebuttals addressed your concerns or clarified anything?

---

### Author Response · Authors · 2025-12-02

We thank all reviewers for their feedback and time, we summarize their reviews and our interactions: **xjHV** found the work rigorous, well written, and the problem and motivation clearly stated. **Cemk** found the work careful, the problem important and principled. **vAd1** found the problem interesting and novel, they appreciated the contrast to prior work and found formal languages to be a good fit for studying the problem. **poG2** found the work rigorous, original, elegant, being methodologically and empirically strong, intuitive and establishing a foundation for future studies on causality.

—-

**xjHV (8)** did not give direct weaknesses but raised two clarifying questions which we answered.

**Cemk (4)** points out two weaknesses. First, that the paper’s presentation of the theoretical contribution in section 4 and the Appendix may overshadow the intuition. We answered the reviewer, making the point that the contribution is in many ways inherently technical and explained our choice of terminology. The second weakness has to do with their understanding of the difference between scarcity and complexity—we formalize this in our second answer to the reviewer pointing out that scarcity is the natural frequency in the data, while complexity is the number of samples needed to reach some low $\epsilon >0 $ error. We will clarify this in the updated manuscript.

**vAd1 (2)** points to the complexity of the theoretical work while acknowledging their **insufficient mathematical background** and inability to provide feedback on the accuracy of the work. They say they did not understand some takeaways from the experiments, we will update the manuscript to align with the answers given to the reviewer. Contrast this with the understanding of **xjHV** who understands the key point of the figures that “intervention results generally differ from observational results”, justifying our methodology.

**poG2 (4 $\to$ 6)** The reviewer commented on the scale of the empirical setup (formal languages, multiple training runs, not large scale models, comparison to non-formal work) which we answered and they acknowledged clarified their concerns and raised their score.

---

### Meta-Review · Area_Chair_6bxr · 2026-01-16

**Summary:**

The paper proposes a causal framework to disentangle scarcity (low data frequency) from complexity (intrinsic learnability) in language models using Probabilistic Finite-State Automata (PFSAs). While reviewers generally agreed that the problem is important and the motivation is compelling, the submission faces significant hurdles regarding clarity and conceptual grounding.

The primary concerns center on the "heavy" mathematical formalism in Section 4, which some reviewers felt overshadowed the intuition. Furthermore, a critical disagreement persists regarding the definitions of scarcity and complexity; specifically, whether "complexity" as defined here is simply a repackaging of sample complexity. While the authors provided detailed technical responses, the lack of consensus on the paper’s fundamental takeaways and its narrow empirical scope (limited to synthetic PFSAs) weighs against a strong acceptance.

**Reviewer Concerns:**

Addressed by Rebuttal

- The authors addressed concerns of Reviewer poG2 regarding the scale of the experiments (using formal languages vs. large-scale models), which led Reviewer poG2 to raise their score from a 4 to a 6.

- The authors explained the necessity of the "binning semiring" for efficient, controlled sampling, justifying the technical depth required for the causal intervention.

Outstanding Concerns

- Reviewer Cemk remains unconvinced by the distinction between scarcity and complexity, arguing that the parity example demonstrates sample complexity rather than a new "complexity" category. This indicates a fundamental disagreement on the theoretical contribution.

- Despite the rebuttal, Reviewer vAd1’s concerns regarding the difficulty of connecting the math to the high-level takeaways remain significant. The reviewer felt "entirely lost" in the methodology section.

- While Reviewer poG2 acknowledged the clarity of the rebuttal, the concern that the framework is limited to discrete PFSA settings and may not scale to real-world datasets remains an inherent limitation of the current work.

**Reviewer Scores:**

- Reviewer poG2: 4->6
Explicitly stated that the authors clarified the concerns.
- Reviewer Cemk: 4->4
Explicitly stated that they maintain the score due to lingering conceptual objections.
- Reviewer xjHV: 8->8
Remained highly positive; the rebuttal clarified their questions on the architecture comparison.
- Reviewer vAd1: 2->2
While the authors clarified the figure takeaways, the overall presentation will still be a major weakness.

---

### Decision · Program_Chairs · 2026-01-26

Reject